# Curcumin-Based Nanoparticles: Advancements and Challenges in Tumor Therapy

**DOI:** 10.3390/pharmaceutics17010114

**Published:** 2025-01-15

**Authors:** Hicham Wahnou, Riad El Kebbaj, Bertrand Liagre, Vincent Sol, Youness Limami, Raphaël Emmanuel Duval

**Affiliations:** 1Laboratory of Immunology and Biodiversity, Faculty of Sciences Ain Chock, Hassan II University, B.P2693, Maarif, Casablanca 20100, Morocco; hwwahnou@gmail.com; 2Laboratory of Health Sciences and Technologies, Higher Institute of Health Sciences, Hassan First University of Settat, Settat 26000, Morocco; elkebbajriad@gmail.com; 3Univ. Limoges, LABCiS, UR 22722, F-87000 Limoges, France; bertrand.liagre@unilim.fr (B.L.); vincent.sol@unilim.fr (V.S.); 4Université de Lorraine, F-54000 Nancy, France

**Keywords:** curcumin, nanoparticles, bioavailability, cancer therapy, targeted delivery, apoptosis, tumor treatment

## Abstract

Curcumin, a bioactive compound derived from the rhizome of *Curcuma longa* L., has garnered significant attention for its potent anticancer properties. Despite its promising therapeutic potential, its poor bioavailability, rapid metabolism, and low water solubility hinder curcumin’s clinical application. Nanotechnology offers a viable solution to these challenges by enabling the development of curcumin-based nanoparticles (CNPs) that enhance its bioavailability and therapeutic efficacy. This review provides a comprehensive overview of the recent advancements in the design and synthesis of CNPs for cancer therapy. We discuss various NP formulations, including polymeric, lipid-based, and inorganic nanoparticles, highlighting their role in improving curcumin’s pharmacokinetic and pharmacodynamic profiles. The mechanisms by which CNPs exert anticancer effects, such as inducing apoptosis, inhibiting cell proliferation, and modulating signaling pathways, are explored in details. Furthermore, we examine the preclinical and clinical studies that have demonstrated the efficacy of CNPs in treating different types of tumors, including breast, colorectal, and pancreatic cancers. Finally, the review addresses the current challenges and future perspectives in the clinical translation of CNPs, emphasizing the need for further research to optimize their design for targeted delivery and to enhance their therapeutic outcomes. By synthesizing the latest research, this review underscores the potential of CNPs as a promising avenue for advancing cancer therapy.

## 1. Introduction

Cancer remains a global health burden, with millions of lives lost each year despite advances in conventional therapies such as chemotherapy, radiotherapy, and targeted treatments [1]. The complexity of tumor biology and the limitations of current therapeutic approaches have spurred interest in natural compounds with potential anticancer properties [2]. Among these, curcumin, the bioactive polyphenol extracted from *Curcuma longa* L. (turmeric), has gained widespread recognition for its broad spectrum of biological activities, particularly its anticancer potential (Figure 1) [3]. Curcumin has demonstrated the ability to modulate several key pathways involved in tumor development and progression, such as apoptosis, angiogenesis, and metastasis [3]. It also exerts anti-inflammatory, antioxidant, and immunomodulatory effects, further supporting its therapeutic relevance [4]. However, despite its potential as a multitarget anticancer agent, curcumin’s clinical application has been severely limited due to its poor solubility in water (0.6 µg/mL) [5]. Low bioavailability (1%), in fact, studies have shown that the highest amount of plasma curcumin concentration was 0.051 µg/mL from 12 g curcumin in human, 1.35 µg/mL from 2 g/kg in rat, and 0.22 µg/mL from 1 g/kg in mouse [6].Rapid systemic metabolism, studies indicate that the primary metabolic pathways of curcumin in animals involve phase I reduction reactions, resulting in the formation of metabolites such as dihydrocurcumin, tetrahydrocurcumin, octahydrocurcumin, and hexahydrocurcumin [7]. and poor cellular uptake [8,9]. These pharmacokinetic hurdles have prompted extensive research into novel delivery systems aimed at improving curcumin’s therapeutic potential in oncology. Nanotechnology has emerged as a promising solution to these challenges [10,11,12]. Indeed, curcumin-based nanoparticles (CNPs), including organic, inorganic and carbon-based NPs, have been able to significantly enhance its bioavailability and stability [10]. CNPs offer controlled and targeted delivery to tumor cells, minimize off-target effects, and improve therapeutic efficacy, making them an attractive option for cancer treatment [13].

This review aims to provide a comprehensive overview of the recent advancements of CNPs for tumor therapy. It highlights the mechanisms by which CNPs exert their anticancer effects and evaluates preclinical and clinical studies that underscore their potential. Additionally, the challenges in translating CNPs to clinical practice are discussed, along with future research directions for optimizing their design and improving their therapeutic outcomes.

## 2. Curcumin’s Mechanism of Action in Cancer Therapy

Curcumin, has garnered attention for its broad range of anticancer effects. Indeed, curcumin acts through various molecular mechanisms, which interfere with cancer progression by targeting multiple signaling pathways (Figure 2).

### 2.1. Reulation of Inflammation and Immune Modulation

Chronic inflammation is a major contributor to the initiation and progression of cancer. Tumor-promoting inflammation leads to the release of pro-inflammatory cytokines, chemokines, and reactive oxygen species (ROS), creating a microenvironment that favors cancer development [4,14,15,16,17]. Curcumin has been extensively studied for its anti-inflammatory properties, particularly its ability to modulate key inflammatory pathways involved in cancer progression (Figure 2A).

One of the key targets of curcumin is the nuclear factor kappa-light-chain-enhancer of activated B cells (NF-κB), a transcription factor that regulates genes responsible for inflammation and cell survival [18]. In cancerous cells, NF-κB is often constitutively activated, promoting the expression of pro-inflammatory cytokines such as tumor necrosis factor-alpha (TNF-α), interleukin-1β (IL-1β), and cyclooxygenase-2 (COX-2), which contribute to tumor growth and metastasis [19]. Curcumin inhibits NF-κB activation, thereby reducing the production of these cytokines and inflammatory mediators, creating a less favorable environment for cancer cell survival [18].

Additionally, curcumin reduces ROS levels by acting as a scavenger of free radicals. Elevated ROS levels are associated with DNA damage, mutations, and cancer progression [4]. By neutralizing ROS, curcumin protects cells from oxidative stress and inhibits cancer initiation and progression (Figure 2A).

Curcumin also affects immune modulation by enhancing the activity of various immune cells, including T cells, B cells, macrophages, and natural killer cells [20]. It modulates the production of cytokines, such as interferon-gamma (IFN-γ), which helps in stimulating the immune response against cancer cells [21]. Furthermore, curcumin has been found to inhibit the maturation of dendritic cells and downregulate co-stimulatory molecules like CD80 and CD86, reducing the immune system’s overactivation, which can be beneficial in tumor-associated immune suppression [22].

### 2.2. Effects on Cancer Stem Cells

Cancer stem cells (CSCs) are a subpopulation of cells within tumors that possess self-renewal and differentiation capabilities, often leading to resistance against conventional therapies and playing critical roles in tumor initiation, metastasis, and recurrence [23]. Curcumin has been shown to modulate CSC key signaling pathways for their maintenance and function [24,25]. For instance, curcumin disrupts the Wnt/β-catenin pathway, preventing β-catenin from accumulating in the nucleus, which inhibits the transcription of Wnt target genes that enhance CSC self-renewal [26]. Additionally, curcumin affects the Notch signaling pathway, leading to a decrease in CSC properties and promoting differentiation into non-tumorigenic cells [27]. It also inhibits the signal transducer and activator of transcription 3 (STAT3) pathway, reducing the expression of stemness-related genes, and impacts the Hedgehog pathway, which is crucial for stem cell maintenance [28]. Moreover, curcumin induces apoptosis in CSCs by modulating apoptosis-regulating proteins, enhancing their sensitivity to traditional chemotherapy [29]. By targeting and reducing the population of CSCs, curcumin effectively prevents tumor recurrence, making it a valuable therapeutic agent in cancer treatment (Figure 2B).

### 2.3. Induction of Apoptosis and Cell Cycle Arrest

One of the most important mechanisms by which curcumin exerts its anticancer effects is through the induction of apoptosis, a process of programmed cell death that is often dysregulated in cancer [30]. Apoptosis is regulated by a balance between pro-apoptotic and anti-apoptotic proteins [30]. In many types of cancer, anti-apoptotic proteins, such as Bcl-2, are overexpressed, leading to resistance to cell death [30]. Curcumin shifts this balance by downregulating anti-apoptotic proteins (e.g., Bcl-2, Bcl-xL) and upregulating pro-apoptotic proteins (e.g., Bax, Bak) [30]. This results in the activation of the mitochondrial apoptotic pathway (Figure 2C). Curcumin induces mitochondrial dysfunction by causing the release of cytochrome c, which activates pro-apoptotic caspases, the proteases responsible for dismantling the cell during apoptosis, in particular caspase-3 [31].

Curcumin also induces cell cycle arrest at various checkpoints, particularly at the G2/M phase, which is critical for cell division [32]. Cancer cells often bypass these checkpoints due to mutations in cell cycle regulators like cyclin-dependent kinases (CDKs) [33]. Curcumin inhibits the activity of CDKs by downregulating cyclin D1 and CDK4/6, thereby halting the cell cycle and preventing cancer cells from proliferating [34]. This blockage of the cell cycle allows the cells to either repair damages or proceed to apoptosis, both reducing tumor growth and proliferation [34] (Figure 2D). Additionally, curcumin has been shown to activate the tumor suppressor protein p53, which plays a central role in regulating apoptosis and the cell cycle [35]. In many cancers, p53 is either mutated or inactivated, allowing cancer cells to proliferate uncontrollably. Curcumin reactivates p53, leading to the transcription of pro-apoptotic genes and the induction of cell cycle arrest [35] (Figure 2D).

### 2.4. Inhibition of Angiogenesis and Metastasis

Angiogenesis, the formation of new blood vessels, is crucial for tumor growth, as it supplies the cancer cells with oxygen and nutrients needed for rapid proliferation [36]. Curcumin has been shown to inhibit angiogenesis through various mechanisms, particularly by downregulating vascular endothelial growth factor (VEGF), one of the primary factors that promotes blood vessel formation in tumors [37]. VEGF is often overexpressed in tumors, leading to excessive angiogenesis, which supports tumor expansion and metastasis [38]. Curcumin decreases VEGF expression in cancer cells, thereby limiting the tumor’s ability to form new blood vessels and sustain its growth [37] (Figure 2E).

In addition to VEGF inhibition, curcumin targets other angiogenesis-related pathways, such as the hypoxia-inducible factor-1 alpha (HIF-1α) pathway, which is activated in low-oxygen (hypoxic) conditions commonly found in solid tumors [39]. HIF-1α stimulates the transcription of angiogenesis-related genes, including VEGF [37,39]. By suppressing HIF-1α activity, curcumin further impedes the angiogenic process, creating an environment that is less conducive to tumor growth.

Curcumin also plays a significant role in inhibiting metastasis, the process by which cancer cells spread from the primary tumor to distant organs [40]. It disrupts metastasis by inhibiting the activity of matrix metalloproteinases (MMPs), particularly MMP-2 and MMP-9, which degrade the extracellular matrix (ECM) and allow cancer cells to invade surrounding tissues and enter the bloodstream [40]. By downregulating MMP expression, curcumin impedes the ability of cancer cells to invade and metastasize (Figure 2F).

Furthermore, curcumin modulates the epithelial-mesenchymal transition (EMT), a process by which epithelial cancer cells acquire mesenchymal, invasive properties [41]. EMT is essential for metastasis, as it enhances the motility of cancer cells [42]. In addition, curcumin inhibits key signaling pathways, such as the Wnt/β-catenin and TGF-β pathways, which are critical in promoting EMT [43]. By inhibiting these pathways, curcumin prevents cancer cells from detaching from the primary tumor and invading distant tissues.

Through its combined effects on angiogenesis and metastasis, curcumin not only reduces tumor growth but also minimizes the risk of cancer spreading to other organs, making it a potent anticancer agent with multifaceted action.

## 3. Challenges of Curcumin in Cancer Therapy

Curcumin, has gained considerable attention for its potential therapeutic effects against various types of cancer. Despite its promising anticancer properties, several challenges hinder the clinical application of curcumin in cancer therapy. These challenges primarily revolve around its poor bioavailability, rapid metabolism, low solubility, and the complexity of its pharmacokinetics (Figure 3). Understanding these limitations is crucial for developing effective strategies to enhance the therapeutic potential of curcumin in oncology.

### 3.1. Poor Bioavailability

One of the most significant challenges associated with curcumin is its poor bioavailability [44]. Curcumin’s bioavailability is markedly low due to several factors, including its rapid metabolism, extensive first-pass effect, and limited absorption in the gastrointestinal tract [44].

Studies have shown that after oral administration, curcumin is quickly metabolized by the liver and intestines, leading to the formation of various metabolites that have reduced biological activity compared to the parent compound [45]. For example, curcumin is primarily conjugated to glucuronides and sulfates, which are excreted in urine, significantly limiting the amount of active curcumin that reaches the bloodstream and target tissues [46]. This rapid metabolism results in low plasma concentrations, which do not achieve the levels necessary for effective anticancer activity (Figure 3A).

Moreover, curcumin’s hydrophobic nature contributes to its poor bioavailability with a saturation solubility in water of 0.6 µg/mL [5]. Hence, curcumin is poorly soluble in water, which limits its absorption in the intestinal tract, a critical step for oral administration. The low solubility results in limited dissolution in the gastrointestinal fluids, thereby reducing the amount of curcumin that can be absorbed through the intestinal lining [44].

### 3.2. Rapid Metabolism and Elimination

In addition to its poor bioavailability, curcumin undergoes rapid metabolism and elimination from the body, further complicating its clinical application. Curcumin is subjected to extensive biotransformation in the liver, where it is rapidly metabolized into various conjugated forms. The metabolites, which are often less active than curcumin itself, are excreted primarily through urine and feces [45] (Figure 3A).

The rapid metabolism of curcumin can significantly shorten its half-life in the bloodstream, which affects its therapeutic efficacy [45]. For instance, studies have reported that the plasma half-life of curcumin is less than one hour following oral administration, leading to a swift decline in curcumin levels in circulation [47]. This short plasma half-life necessitates frequent dosing, which can be impractical for patients and may lead to inconsistent therapeutic outcomes. Moreover, the rapid elimination of curcumin limits its potential for sustained action against cancer cells.

### 3.3. Dosing and Administration Challenges

The optimal dosing regimen for curcumin remains uncertain due to its variable pharmacokinetics. Existing clinical trials have employed a wide range of doses, from low daily doses to high doses administered in divided regimens, reflecting a lack of consensus on effective dosing strategies [48]. This variability in dosing can lead to inconsistent therapeutic outcomes and complicates the interpretation of clinical results.

Furthermore, although curcumin has been reported to induce DNA damage both in vitro and in vivo due to its pro-oxidant effects [49,50], clinical evidence indicates a favorable safety profile. A Phase 1 human trial involving 25 participants demonstrated that daily doses of up to 8000 mg of curcumin for three months caused no detectable toxicity. Moreover, five additional human trials utilizing doses between 1125 and 2500 mg per day also confirmed its safety [51].

Moreover, the administration route also influences the pharmacokinetics of curcumin. Oral administration is the most common method used in clinical studies; however, its bioavailability issues necessitate high doses to achieve therapeutic effects [48]. Alternative routes, such as intravenous administration, offer the potential for higher bioavailability but require suitable formulations that overcome curcumin’s solubility challenges (Figure 3B).

### 3.4. Interactions with Other Medications

Curcumin has been shown to interact with various drugs, which can complicate its clinical application, particularly in cancer patients who are often on multiple medications [52]. Curcumin can particularly interact with CYP3A4 and CYP2C9, which are involved in the metabolism of many chemotherapeutic agents, anticoagulants, and other commonly used drugs [53]. By inhibiting these enzymes, curcumin can increase the plasma concentrations of co-administered drugs, potentially leading to enhanced effects or toxicity. This poses a significant challenge in cancer therapy, where patients may already be taking multiple medications with narrow therapeutic windows [53].

For example, curcumin’s interaction with warfarin and other anticoagulants can enhance their blood-thinning effects, increasing the risk of bleeding [54]. Similarly, curcumin has been found to interfere with chemotherapeutic agents such as paclitaxel and doxorubicin (DOX), potentially altering their pharmacokinetics and efficacy [55,56]. These interactions necessitate careful consideration of drug combinations and monitoring during curcumin-based therapy (Figure 3C).

Moreover, curcumin’s effects on P-glycoprotein (P-gp), a drug efflux pump, further complicates its interactions with other drugs [57]. P-gp plays a critical role in the absorption and distribution of many chemotherapeutic agents [58]. Curcumin has been shown to inhibit P-gp, which can lead to increased intracellular concentrations of certain drugs, enhancing their efficacy but also raising the risk of toxicity [59].

### 3.5. Stability and Degradation

Curcumin’s chemical stability is another key challenge in its clinical application. The compound is highly unstable under physiological conditions, particularly at neutral or alkaline pH, where it undergoes rapid degradation [60]. Curcumin is prone to degradation into various byproducts, some of which have lower biological activity [60]. For example, under physiological conditions, curcumin undergoes degradation, resulting in the formation of several byproducts, including trans-6-(4′-hydroxy-3′-methoxyphenyl)-2,4-dioxo-5-hexenal, ferulic aldehyde, ferulic acid, feruloyl methane, and vanillin [61].

Additionally, curcumin is metabolized in both humans and rodents via conjugation and reduction pathways [61]. Following oral administration, curcumin is conjugated, leading to the production of curcumin glucuronide and curcumin sulfates, while intraperitoneal or systemic administration results in its reduction to form tetrahydrocurcumin, hexahydrocurcumin, and octahydrocurcumin [61].

Additionally, the light-sensitive nature of curcumin adds another layer of complexity [62]. Exposure to light, especially UV light, can further accelerate its degradation, which poses challenges in packaging and storage [62] (Figure 3D).

Furthermore, curcumin is sensitive to both temperature and pH. While previous studies have indicated that curcuminoids remain stable when exposed to heat (80 °C for 2 h), they become increasingly susceptible to degradation under acidic or alkaline conditions. This instability is likely influenced by the chemical structure of curcuminoids, including the diketone moiety, methoxy groups, and hydroxyl groups, which contribute to their unique degradation behavior [63,64]. Notably, in non-coated emulsions, 26.1% of curcumin was degraded during autoclaving, highlighting its vulnerability to thermal processing [65].

Overcoming these stability issues is crucial for the development of reliable and effective curcumin-based cancer therapies.

## 4. Nanotechnology for Curcumin Delivery

To overcome these challenges, nanotechnology has emerged as a promising approach to enhance the delivery and efficacy of curcumin in cancer therapy. Researchers can improve its pharmacokinetics, enable targeted delivery, and enhance its therapeutic action against tumors through nanoformulation. CNPs are broadly classified into three categories: organic, inorganic, and carbon-based nanoparticles (Figure 4). This section explores some various types of CNPs and the mechanisms through which they improve curcumin’s anticancer effects.

### 4.1. Types of Curcumin-Based Nanoparticles

#### 4.1.1. Organic NPs

Organic NPs are composed of biocompatible and biodegradable materials, making them ideal carriers for curcumin. These systems include liposomes, dendrimers, polymers, micelles, emulsions, and nanogels (Figure 4).

Liposomes

Liposomes are spherical vesicles formed by phospholipid bilayers that encapsulate hydrophobic drugs like curcumin within their lipid core [66]. They protect curcumin from degradation while improving solubility and facilitating targeted delivery. Liposomes have been extensively used in cancer therapy due to their ability to accumulate in tumors via enhanced permeability and retention (EPR) effects [66]. Additionally, their surface can be modified with targeting ligands to enhance specificity for cancer cells [66].

2.Dendrimers

Dendrimers are highly branched, tree-like polymers that offer a large surface area for drug conjugation [67]. These NPs provide a high drug-loading capacity and the ability to release curcumin in a controlled manner [67]. Functionalization of dendrimers can enhance their biocompatibility and targeting potential, making them suitable for systemic and localized curcumin delivery [67].

3.Polymeric NPs

Polymers like polylactic-co-glycolic acid (PLGA), chitosan, and polyethylene glycol (PEG) have been widely used to fabricate NPs for curcumin delivery. These systems encapsulate curcumin within a polymer matrix, protecting it from rapid degradation and metabolism [68]. Controlled and sustained drug release ensures prolonged therapeutic effects. Surface modification with targeting moieties, such as folic acid, further enhances their ability to bind to specific cancer cells [69].

4.Micelles

Micelles are self-assembled structures formed by amphiphilic molecules. Their hydrophobic core traps curcumin, while the hydrophilic shell ensures stability in aqueous environments. Micelles are particularly advantageous for oral delivery, as they overcome curcumin’s poor solubility in gastrointestinal fluids [70,71]. Additionally, they protect curcumin from enzymatic degradation, enhancing its systemic absorption [70,72].

5.Emulsions and Nanogels

Emulsions are thermodynamically stable systems consisting of two immiscible liquids, such as oil and water, used to solubilize curcumin [73]. Nanogels, on the other hand, are three-dimensional polymeric networks that provide a soft, flexible matrix for drug encapsulation [74]. Both systems are suitable for delivering curcumin in a controlled and sustained manner, improving its therapeutic efficacy [73].

6.Exosomes

Exosomes are naturally occurring extracellular vesicles secreted by cells, serving as efficient carriers for delivering bioactive molecules, including proteins, lipids, and nucleic acids, to target cells [75]. Their unique properties, such as biocompatibility, low immunogenicity, and the ability to cross biological barriers, make them a highly promising delivery system for therapeutic agents [75]. Exosomes can be engineered or loaded with drugs, offering targeted and efficient delivery while minimizing off-target effects and systemic toxicity [76]. By encapsulating curcumin in exosomes, its stability and solubility are significantly enhanced, ensuring effective delivery to target cells or tissues [77]. This exosome-curcumin system not only protects curcumin from enzymatic degradation but also facilitates its uptake by target cells, increasing its therapeutic efficacy and reducing the required dosage [76,77].

#### 4.1.2. Inorganic NPs

Inorganic NPs are composed of non-organic materials such as gold, silica, or iron oxides. These NPs provide unique properties like imaging capabilities, thermal stability, and functional versatility, making them suitable for advanced therapeutic applications (Figure 4).

Gold NPs

Gold NPs are widely explored for their ability to deliver curcumin while simultaneously enabling imaging through surface plasmon resonance (SPR) [78]. They are ideal for theranostics, combining therapeutic delivery and real-time tumor imaging [78]. The functionalization of AuNPs with targeting ligands enhances their specificity for cancer cells, reducing systemic toxicity [79].

2.Silica NPs

Silica NPs possess a high surface area and large pore volume, allowing them to load significant amounts of curcumin [80]. These NPs offer excellent stability and controlled release profiles, ensuring sustained therapeutic action [81]. Their versatility allows for surface modification to improve biocompatibility and targeting efficiency [81].

3.Iron Oxide NPs

Magnetic iron oxide NPs enable the targeted delivery of curcumin through the application of an external magnetic field [82]. These NPs are particularly useful in precision medicine, as they allow curcumin to be concentrated at tumor sites, minimizing off-target effects [83]. Additionally, their magnetic properties can be utilized for imaging applications.

4.Quantum Dots

Quantum dots (QDs) are nanoscale semiconductors with unique optical properties [84]. They provide imaging capabilities alongside drug delivery, making them an excellent tool for theranostics [84]. Curcumin-loaded QDs are highly promising for cancer treatment, as they allow for real-time tracking of NP distribution and therapeutic efficacy [85].

#### 4.1.3. Carbon-Based NPs

Carbon-based NPs offer exceptional drug-loading capacity, biocompatibility, and versatility [86,87,88,89]. These systems include fullerenes, graphene, and carbon nanotubes, each with unique advantages for curcumin delivery [90] (Figure 4).

5.Fullerenes

Fullerenes are spherical carbon nanostructures with a hollow interior, providing an excellent platform for encapsulating curcumin [91]. They protect curcumin from degradation and ensure sustained release. Fullerenes can also be functionalized with targeting ligands to enhance their specificity for diseased tissues [91].

6.Graphene and graphene oxide

Graphene and its oxidized derivative, graphene oxide, are two-dimensional carbon materials with a high surface area and excellent mechanical properties [92]. These NPs are capable of adsorbing large amounts of curcumin onto their surface [93]. Functionalization with biocompatible polymers or targeting ligands enhances their ability to cross biological barriers, such as the blood-brain barrier, and deliver curcumin to specific tissues [93].

7.Carbon nanotubes

Carbon nanotubes (CNTs) are cylindrical nanostructures that provide a unique platform for curcumin delivery. Their high aspect ratio and hollow core allow for efficient drug encapsulation and release [89]. Additionally, CNTs can pass through biological barriers and deliver curcumin directly to target cells [94]. Functionalized CNTs with biocompatible coatings minimize toxicity and improve their therapeutic index [94].

### 4.2. Mechanisms of CNPs in Tumor Therapy

CNPs enhance the anticancer efficacy of curcumin through multiple mechanisms, including improved bioavailability, targeted delivery, and the modulation of key molecular pathways involved in cancer progression. Therefore, researchers aim to optimize CNPs pharmacokinetics and therapeutic profile, making them more effective in combating tumor growth and metastasis.

#### 4.2.1. Enhanced Bioavailability and Prolonged Circulation

One of the primary mechanisms by which CNPs improve curcumin’s therapeutic efficacy is by enhancing its bioavailability and circulation time in the body [13]. NPs protect curcumin from rapid degradation and metabolism, allowing for a more sustained release into the bloodstream [95]. The encapsulation of curcumin in polymeric, lipid-based, or inorganic NPs prevents its premature breakdown, increasing the concentration of bioactive curcumin that reaches tumor tissues [13]. Additionally, nanoparticle coatings, such as polyethylene glycol (PEG), can extend the circulation time of CNPs by reducing clearance by the reticuloendothelial system (RES), ensuring that curcumin remains in circulation long enough to exert its therapeutic effects on cancer cells [96].

#### 4.2.2. Targeted Delivery and Tumor Accumulation

CNPs offer the advantage of targeted delivery to tumor sites, enhancing curcumin’s efficacy while minimizing damage to healthy tissues [97]. NPs can be engineered to exploit the enhanced permeability and retention (EPR) effect, a phenomenon in which NPs preferentially accumulate in tumor tissues due to the leaky vasculature surrounding tumors [98]. This passive targeting allows CNPs to deliver high concentrations of curcumin directly to cancer cells, increasing its therapeutic efficacy [97].

In addition to passive targeting, NPs can be functionalized with specific ligands, such as antibodies, peptides, or small molecules, that bind to receptors overexpressed on cancer cells [99]. This active targeting mechanism allows for even greater precision in delivering curcumin to tumor sites. For example, NPs coated with folic acid can specifically target cancer cells that overexpress folate receptors, improving curcumin’s selective uptake by malignant cells [100].

#### 4.2.3. Induction of Apoptosis and Inhibition of Proliferation

CNPs enhance curcumin’s ability to induce apoptosis and inhibit cancer cell proliferation [101]. Once delivered to the tumor site, curcumin encapsulated in NPs can effectively interact with key molecular pathways that regulate cell survival and growth. Curcumin targets several signaling pathways involved in cancer progression, including the NF-κB pathway, the PI3K/Akt pathway, and the Wnt/β-catenin pathway.

By inhibiting the NF-κB pathway, curcumin suppresses the expression of anti-apoptotic proteins and inflammatory cytokines, promoting cancer cell death [18]. Additionally, curcumin’s inhibition of the PI3K/Akt pathway leads to decreased cell survival and increased sensitivity to apoptosis [102]. The downregulation of the Wnt/β-catenin pathway by curcumin prevents cancer cell proliferation and tumor growth [43]. The encapsulation of curcumin in NPs enhances its interaction with these pathways by ensuring higher concentrations of curcumin reach the tumor site and remain active for extended periods, making the therapeutic effects more pronounced [13]. Moreover, liposomal curcumin has been shown to suppress the expression of NF-κB, TNF-α, and COX-2, a key targets involved in inflammation and cancer progression [103], an effect similar to free curcumin [104].

#### 4.2.4. Overcoming Multidrug Resistance

One of the major obstacles in cancer therapy is the development of multidrug resistance (MDR) by cancer cells, which often leads to treatment failure [105]. MDR occurs when cancer cells become resistant to a variety of chemotherapeutic drugs, primarily through the overexpression of drug efflux pumps, such as P-gp, which actively remove drugs from cancer cells, reducing their intracellular concentrations and efficacy [106].

Curcumin has shown potential in overcoming MDR by inhibiting the activity of these efflux pumps [59]. When delivered via NPs, curcumin is protected from immediate efflux by these pumps, allowing it to accumulate in resistant cancer cells and exert its therapeutic effects [107]. Furthermore, curcumin’s ability to modulate the expression of P-gp and other MDR-related proteins helps restore the sensitivity of cancer cells to chemotherapy [108,109]. By combining CNPs with conventional chemotherapeutic agents, it is possible to enhance drug retention within cancer cells and reverse drug resistance, improving overall treatment outcomes.

#### 4.2.5. Reduced Systemic Toxicity

One of the key benefits of using CNPs for curcumin delivery is the reduction of systemic toxicity [97]. Conventional cancer therapies, such as chemotherapy and radiotherapy, often lead to significant side effects due to the damages they cause to healthy tissues [110]. NPs help mitigate these side effects by enhancing the selectivity of curcumin delivery to cancer cells while sparing normal cells [13]. The targeted delivery of curcumin, achieved through both passive (EPR effect) and active targeting mechanisms, ensures that higher concentrations of the drug are localized in the tumor, reducing the exposure of healthy tissues to curcumin and minimizing off-target effects [111].

In addition, the controlled release properties of NPs allow for a gradual and sustained release of curcumin, which helps maintain therapeutic levels in the tumor environment over extended periods without causing spikes in drug concentration that could lead to toxicity [112]. This controlled delivery system enhances patient safety and tolerability, making CNPs a more feasible option for long-term cancer therapy.

#### 4.2.6. Synergistic Effects with Other Therapies

CNPs have demonstrated the ability to work synergistically with other cancer therapies, enhancing the overall efficacy of treatment. Curcumin’s natural anti-inflammatory, antioxidant, and antiproliferative properties complement the mechanisms of traditional chemotherapeutic agents, radiation therapy, and immunotherapy [113,114]. When curcumin is delivered via NPs, these synergistic effects are amplified due to the enhanced bioavailability and targeted delivery.

For instance, curcumin can sensitize cancer cells to chemotherapeutic agents by inhibiting survival pathways, such as the PI3K/Akt and NF-κB pathways, which are often upregulated in resistant cancer cells [24]. Similarly, curcumin can enhance the efficacy of radiation therapy by reducing the DNA repair capacity of cancer cells, making them more susceptible to radiation-induced damage [115]. The combination of CNPs with conventional therapies has shown promise in preclinical studies, leading to improved tumor reduction and patient outcomes while minimizing the doses of chemotherapy or radiation required, thus reducing side effects [116].

## 5. Preclinical Studies of CNPs

Preclinical investigations have shown that CNPs can significantly enhance the therapeutic efficacy of curcumin against various cancer types by improving its bioavailability, solubility, and cellular uptake. This detailed section explores preclinical studies related to the anticancer activity of CNPs in various types of cancers (Table 1, Figure 5 and Figure 6).

### 5.1. Liver Cancer

One notable formulation is mucoadhesive curcumin nanotherapy, which has shown promising results in in vitro studies using HuH7 and HepG2 liver cancer cell lines [117]. The mucoadhesive properties allow for prolonged retention at the tumor site, increasing the local concentration of curcumin. This formulation demonstrated a dose-dependent reduction in cell viability (IC_50_ of 2.5–20 μM) and a marked increase in apoptosis, and necrosis [117]. Furthermore, flow cytometry and cytotoxicity assays demonstrate the potential of encapsulated curcumin to inhibit liver cancer cell proliferation [117]. Such mechanisms underscore its potential to effectively target and destroy liver cancer cells.

In addition, targeted dendrimeric curcumin represents a more refined approach, improving curcumin’s targeting ability. Studies on HuH7 and Hepa1-6 cell lines revealed that this formulation not only decreased cell viability (IC_50_ of 5–50 μM) but also induced cell cycle arrest at the G2/M phase, thereby inhibiting cell proliferation [67]. Furthermore, it significantly increased ROS levels while depleting intracellular ATP and glutathione, which are essential for cancer cell survival [67]. In in vivo models using Hepa1-6 xenografts, targeted dendrimeric curcumin reduced tumor growth and extended survival rates in mice (with a dose of 0.5 mg/25 g), further validating its potential as a liver cancer therapy [67].

Another innovative approach involves curcumin-loaded nanoechinus, which has been tested on HepG2-bearing mice [118]. While specific IC_50_ data is not available, this formulation exhibited a strong tumor-suppressing effect by reducing both tumor volume and weight [118]. Additionally, when tested on HepG2 cell lines, nanoechinus curcumin triggered significant cytotoxicity and green fluorescence signals indicative of cellular damage, with an IC_50_ of approximately 10 μg/mL [118].

Curcumin carbon nanodots also show remarkable anticancer potential. These NPs, with an IC_50_ range of 0.1–3.2 mg/mL, enhanced curcumin’s ability to induce apoptosis and reduce cell viability in liver cancer cells [119]. The small size and high surface area of carbon nanodots improve curcumin delivery to cancer cells, further amplifying its therapeutic effects.

Other nanoformulations, such as iron oxide (Fe_3_O_4_)@curcumin-loaded layered double hydroxide/polydopamine (LDH/PDA) and curcumin-loaded KGM-gAH8 micelles, have demonstrated similar efficacy, with a focus on enhancing cytotoxicity and reducing cell viability. For instance, Fe_3_O_4_@curcumin-LDH/PDA reduced cell viability at concentrations of 6.125–200 μg/mL [120], while KGM-gAH8 micelles were effective at doses ranging from 2–266 μg/mL, offering a versatile platform for curcumin delivery [86].

Lastly, cisplatin-curcumin coloaded liposomes provide a synergistic approach to liver cancer treatment by combining curcumin with the chemotherapy drug cisplatin. This formulation has been shown to downregulate the expression of Sp1 and Bcl-2, proteins that promote cancer cell survival, while increasing ROS production and activating apoptotic pathways involving p53, caspase-3, and Bax [121]. In in vivo models using HepG2 and H22 xenografts, cisplatin-curcumin coloaded liposomes effectively reduced tumor growth and improved survival rates at a dose of 9.8 mg/kg [121].

### 5.2. Bladder Cancer

Bladder cancer is another malignancy where CNPs have shown significant therapeutic potential. Several nanoparticle systems, such as dendrosomes, halloysite nanotubes-grafted chitosan (HNTsg-CS) NPs, and copper NPs, have demonstrated remarkable efficacy against bladder cancer cell lines by inducing apoptosis, cell cycle arrest, and inhibiting key oncogenic pathways.

Dendrosomes, used in the 5637 bladder cancer cell line, are NP carriers that enhance the bioavailability of curcumin. In in vitro studies, dendrosomes loaded with curcumin induced cell death with a time and dose-dependent manner in the 5637 cell line with an IC_50_ of 17.5 µM [122]. The mechanism of action involves inhibiting key stemness markers like Oct4 (Octamer-binding transcription factor 4), SOX2 (Octamer-binding transcription factor 4), and NANOG, which are crucial for cancer cell proliferation and resistance [122]. Moreover, dendrosomes induce cell cycle arrest, preventing the uncontrolled proliferation of bladder cancer cells [122]. This highlights their potential in targeting both cancer cell growth and stem cell-like characteristics, which are often implicated in therapy resistance and recurrence [122].

Another promising approach involves HNTsg-CS NPs (halloysite nanotubes functionalized with chitosan and curcumin), tested on the EJ-1 bladder cancer cell line. This formulation has demonstrated an IC_50_ of 5.3 µM, significantly reducing cell viability and inducing cell death [123]. HNTsg-CS NPs also trigger cell cycle arrest, further inhibiting cancer cell proliferation. The combination of nanotubes with curcumin enhances the compound’s stability and allows for a controlled release, improving its anti-cancer efficacy [123].

Copper NPs loaded with curcumin have shown potent anti-cancer effects in the TCCSUP bladder cancer cell line, with an IC_50_ of 290 µg/mL [124]. The mechanism involves multiple pathways, including the activation of the p53 signaling pathway [35]. This leads to increased expression of pro-apoptotic proteins like Bax and cleaved caspase-8, and the downregulation of anti-apoptotic proteins like Bcl-2 [124]. In addition to promoting apoptosis, copper NPs also inhibit the STAT3 signaling pathway, which is often upregulated in cancer and plays a crucial role in cell survival and proliferation [124]. Furthermore, these NPs suppress colony formation, a key feature of cancer cells, thereby reducing the ability of bladder cancer cells to grow and spread [124].

### 5.3. Melanoma

CNPs, often combined with other bioactive compounds such as chrysin, have emerged as promising therapeutic strategies for melanoma, an aggressive form of skin cancer. These nanoformulations enhance curcumin’s bioavailability and anticancer activity, making it more effective against melanoma cells by reducing cell viability, promoting cytotoxicity, and inhibiting tumor growth in both cell culture and animal models [125].

One notable example is curcumin and chrysin-loaded NPs tested on B16F10 melanoma cells. Studies by Tavakoli et al. demonstrated that this formulation at concentrations of 5–60 μM significantly reduces cell viability while increasing cytotoxicity [125]. These NPs also inhibit the expression of MMP-2 and MMP-9, enzymes involved in cancer invasion and metastasis. Additionally, they decrease the expression of telomerase reverse transcriptase (TERT), a key component in maintaining cancer cell immortality, and upregulate tissue inhibitors of metalloproteinases (TIMP-1 and TIMP-2), further supporting their anti-metastatic potential [125]. In vivo studies with B16F10-bearing C57BL/6 mice showed that a dose of 30 mg/kg of these NPs effectively reduces tumor growth [125].

Curcumin-loaded NPs have also been used in combination with photothermal therapy (PTT), a technique where NPs are activated by light to kill cancer cells. In a study by Alvi et al., gold liposomal curcumin NPs (Au-Lipos Cur NP) were applied to B16 cells at a concentration of 200 μg/mL, leading to decreased cell growth, reduced viability, and enhanced sensitivity to PTT [126]. The formulation also reduced the expression of proteins such as Hsp70, SLUG, and Mucin, which are associated with cancer cell survival and resistance [126]. In animal models bearing B16 tumors, these NPs showed a significant reduction in tumor growth and increased sensitivity to PTT, highlighting their potential in multimodal cancer therapy [126].

Moreover, another study have shown promising results in both in vitro and in vivo melanoma models using curcumin-loaded poly (propylene carbonate polyol) (PPCP) nanofibrous matrices [127]. In A375 melanoma cells, this formulation decreased cell viability, and in mouse models bearing A375 tumors, it led to reduced tumor growth and increased apoptosis, as evidenced by the downregulation of Ki-67, a marker of cell proliferation [127].

Another curcumin-based formulation, supramolecular nanoassembly of lysozyme and α-lactalbumin SN_LYZ-BLA_-curcumin, also exhibited potent anticancer effects against B16F10 melanoma cells [128]. At concentrations of 40 and 160 μg/mL, this NP significantly decreased cell viability and increased cytotoxicity, suggesting its potential as an effective therapeutic agent for melanoma treatment [128].

Similarly, curcumin-loaded zinc (Zn) and copper (Cu) liposomes have shown strong anticancer effects against B16F10 melanoma cells by enhancing curcumin’s stability and bioavailability [66]. In this study by Zhou et al., Zn liposomes reduced cell viability with an IC_50_ of 4.3 ± 0.6 μg/mL, while Cu liposomes were even more potent with an IC_50_ of 1.3 ± 0.6 μg/mL [66]. These formulations increase cytotoxicity by inducing oxidative stress, with copper boosting ROS production, leading to greater melanoma cell death [66]. These metal-enhanced liposomes present a promising strategy for more effective melanoma treatment.

### 5.4. Colorectal Cancer

Numerous studies have highlighted the therapeutic potential of CNPs in colorectal cancer (CRC), demonstrating significant anti-cancer activity through various mechanisms. Zhang et al. showed that PEGylated curcumin NPs reduced cell viability and increased cytotoxicity in CT26 cells at concentrations ranging from 1 to 40 μM, while in vivo administration of curcumin (10 mg/kg) in CT26-bearing nude mice led to a significant decrease in tumor volume and weight [129]. Similarly, Xie et al. reported that curcumin NPs tested on HCT116 cells (0.01–100 μg/mL) induced cytotoxicity, G2/M cell cycle arrest, and apoptosis [130]. In another approach, using cyclodextrin/carboxymethylcellulose NPs to deliver curcumin, Ntoutoume et al. showed the ability of this nanoformulation to significantly reduce HT29 cell proliferation and viability cells at concentrations of 5 to 50 mM [131]. Further studies demonstrated that a phyto/active gold-fluorescein/chitosan biohybrid reduced viability and increased cytotoxicity in HT29 cells within a 2.5% to 35% concentration range [132]. Dash and Konkimalla showed that encapsulation of curcumin in hydroxypropyl-β-cyclodextrin improved the DOX sensitivity of COLO205 cells (20–60 µM) [133]. In C26 cells, Tefas et al. observed that liposomes co-loaded with curcumin and doxorubicin inhibited cell proliferation at curcumin concentrations of 0.45 to 41.85 mM [134]. Moreover, drug delivery systems incorporating curcumin, either alone or in combination with 5-fluorouracil (5-FU) (1.5–25 μg/mL), significantly reduced cell viability in HCT116 cells [135]. In 2017, Lotfi-Attari et al. reported that curcumin-loaded PLGA/polyethylene glycol (PEG) NPs (12.05 μM) reduced proliferation and increased cytotoxicity in Caco-2 cells, also downregulating the expression of human telomerase reverse transcriptase (hTERT) [136]. In SW480 cells, similar findings were observed with curcumin-loaded PLGA/PEG NPs (3.5–60 μM), leading to decreased proliferation and increased cytotoxicity [137].

Additionally, Sesarman et al. noted that a combination of liposomal curcumin and DOX (20 μM) in C26 cells resulted in decreased proliferation, enhanced cytotoxicity, and downregulation of various cytokines and growth factors [138]. The next year, Al-Ani et al. demonstrated that a curcumin-loaded nanocomposite exhibited cytotoxicity in HT29 and SW948 cells (62.5–1000 μg/mL) [139], while Reimondez-Troitino et al. found that curcumin-loaded protamine nano capsules (2.8, 4.5 μg/mL) in SW480 cells decreased viability and inhibited migration, correlating with increased levels of miR-145 and downregulation of IGF-1R [140].

Furthermore, Sun et al. reported that mesoporous silica NPs loaded with curcumin (10–200 μg/mL) reduced cell viability and increased cytotoxicity in LS174T cells [80]. The same year, Chen et al. indicated that curcumin-loaded NPs (10–100 μg/mL) in Caco-2 cells reduced viability and enhanced cytotoxicity [141]; while Almutairi et al. found that curcumin-loaded NPs (50 μM) in HCT116 cells not only reduced viability but also promoted apoptosis [142]. Xiao et al. highlighted that solid iron-curcumin NPs (0.028–3.6 mg/mL) decreased viability in HT-29 cells and reduced tumor volume in xenograft nude mice [143].

In a more recent study, Ochoa-Sanchez et al. demonstrated that curcumin-resveratrol co-loaded biogenic silica (Cur-Res-BS) exhibited a stronger cytotoxic effect on both cell lines (HCT116 and Caco-2), significantly reducing cell viability, especially in HCT116 cells [144]. The combined treatment was particularly effective, achieving lower IC_50_ values and greater inhibition of cell growth compared to individual treatments. Notably, Cur-Res-BS led to a pronounced reduction in cell viability, with 26.32% at the highest concentration in HCT116 cells after 24 h, highlighting its potential as a potent anticancer agent [144]. Additionally, gene expression analysis revealed that Cur-Res-BS modulated key cancer-related genes, including Wnt-1, CTNNB1, TP53, and Bax, further supporting its anti-proliferative and pro-apoptotic effects in CRC cells [144].

Overall, these studies are merely examples illustrating the remarkable efficiency of various curcumin NPs in reducing cell viability, promoting apoptosis, and enhancing the therapeutic effects against CRC, making them a promising avenue for future research and clinical application.

### 5.5. Ovarian Cancer

Curcumin-loaded NPs have shown significant promise in treating ovarian cancer through various molecular mechanisms. Abtahi et al. investigated curcumin niosomes in A2780s and A2780cp-1 ovarian cancer cells, finding that these NPs increased cytotoxicity and apoptosis. This effect was primarily mediated by the suppression of NF-κB activity and the activation of the tumor suppressor protein p53, demonstrating their potential in triggering programmed cell death [145]. Ghaderi et al. further explored the therapeutic effects of curcumin NPs with the Gemini-curcumin formulation. Their study on OVCAR-3 cells revealed that the treatment effectively decreased cell proliferation and increased apoptosis [146]. Gemini-curcumin also shifted the Bax/Bcl-2 ratio in favor of apoptosis, which underscores its ability to promote cancer cell death by modulating apoptotic pathways [146].

Curcumin-loaded Fe_3_O_4_ NPs significantly decreased cell viability in SKOV-3 cells, indicating the potential of these NPs to suppress ovarian cancer cell growth [147]. Meanwhile, the F68-Cis–Cur formulation demonstrated significant effects in A2780 cells, including decreased cell viability, apoptosis, increased cytotoxicity, and reduced mitochondrial membrane potential (MMPo), suggesting mitochondrial-mediated apoptosis [148]. Steuber et al. explored the effects of curcuminδ-T3 nanoemulsion in OVCAR-8 cells, demonstrating reduced cell viability and enhanced apoptosis. This formulation worked by inhibiting NF-κB activity and promoting caspase 3/7 activation, which are critical for the initiation of apoptosis [149].

Further studies on curcumin NPs have also targeted drug-resistant ovarian cancer cells. Liu et al. showed that curcumin NPs could overcome MDR in A2780 and A2780/ADM cells by downregulating P-gp expression, which is often overexpressed in resistant cancer cells [150]. This breakthrough has significant implications for improving the efficacy of chemotherapy in drug-resistant ovarian cancer.

A recent investigation demonstrated that curcumin-loaded polyethylene glycol-poly (D,L-lactic acid) (PEG-PDLLA NPs) exerted a concentration- and time-dependent suppression of cell proliferation, outperforming free curcumin [151]. As the concentration of PEG-PDLLA NPs increased, both cell scratch-healing and chamber migration abilities were diminished. In comparison to the control group, cells stimulated with lipopolysaccharide (LPS) or overexpressing NF-κB p65 showed markedly higher expression of proteins linked to the NF-κB/PRL-3 signaling pathway, the inflammatory response (TNF-α and IL-6), cell proliferation (cyclin E1 and cyclin A1), and cell migration (N-cadherin and vimentin). Conversely, E-cadherin levels were notably reduced in these groups [151]. Nonetheless, administering higher concentrations of PEG-PDLLA NPs successfully counteracted these alterations.

The combination of curcumin NPs with standard chemotherapy agents has also been evaluated. Sandhiutami et al. investigated the co-administration of curcumin NPs and cisplatin in a 7,12-Dimethylbenz(a)anthracene (DMBA)-induced ovarian cancer rat model [152]. This combined treatment reduced tumor occurrence, volume, and weight while modulating critical pathways, including TGF-β, PI3K, IL-6, and JAK/STAT3. Moreover, the study showed an increase in apoptotic markers such as Bax/Bcl-2 and caspases 3 and 9, further enhancing the apoptotic response in cancer cells [152].

### 5.6. Breast Cancer

Numerous studies have demonstrated the potential anticancer effects of curcumin formulations, especially in breast cancer models. For instance, Shiri et al. observed that dendrosomal curcumin reduced tumor incidence and volume in 4T1 cells bearing mice at doses of 40 and 80 mg/kg, accompanied by a decrease in IL-10, STAT3, and arginase I, while increasing IL-12 and STAT4 [153]. Similarly, Wang et al. showed that curcumin encapsulated in polymeric micelles, combined with DOX, led to reduced cell viability, increased cytotoxicity, and apoptosis in MCF-7 and MCF-7/ADR cells. They further demonstrated that the combination reduced tumor growth and volume in 4T1 cells bearing mice [154].

Additional findings highlight the effectiveness of curcumin conjugates in different cell lines. For example, Sarika et al. found that gum arabic-curcumin micelles reduced cell viability and enhanced cytotoxicity in MCF-7 cells [155]. Similarly, Dey et al. demonstrated that molecular probes@alginate–curcumin-gold NPs (MP@Alg–curcumin AuNPs) significantly reduced cell viability and enhanced cytotoxicity in the same cell line [79]. Cai et al. observed analogous results with curcumin-P123-PAE, reinforcing the broad-spectrum activity of curcumin across different formulations [156]. Liu et al. reported that curcumin loaded in HNTs-g-CS NPs reduced cell viability in MCF-7 cells [123], echoing the results showing that curcumin-loaded PECs in MDA-MB-231 cells induced cytotoxicity, apoptosis, and cell cycle arrest at the G0/G1 phase [157]. Meanwhile, Muthoosamy et al. noted that GP-Cur-Ptx enhanced apoptosis and cytotoxicity in MDA-MB-231 cells by increasing ROS production [207].

Several other studies have also corroborated the anticancer efficacy of curcumin formulations. For instance, Baghbani et al. showed that curcumin-loaded chitosan/perfluorohexane nanodroplets decreased cell viability and increased cytotoxicity in 4T1 cells [158]. Similar results were seen by Baek and Cho who observed enhanced cytotoxicity and decreased P-gp in MCF-7/ADR cells treated with folate conjugated paclitaxel and curcumin/HPCD co-loaded lipid NPs (FPCHN-30) [159]. In T47D cells, Farajzadeh et al. demonstrated that nano-encapsulated metformin-curcumin-PLGA/PEG NPs led to cytotoxicity and human telomerase reverse transcriptase (hTERT) inhibition [160], while Danafar et al. reported that curcumin encapsulated NPs induced apoptosis in SK-BR-3 cells by downregulating Bcl-2 and MMP-9 [161].

Finally, innovative NP formulations continue to demonstrate promising results. Dong et al. observed enhanced cytotoxicity in MCF-7 and MDA-MB-231 cells treated with alendronate-oligoHA-S-S-curcumin (ALN-oHA-S-S-curcumin) [162], while Liu et al. found that icariin and curcumin-loaded polymeric micelles inhibited invasion and reduced tumor growth in MCF-7 cells bearing mice [163]. Borah et al. showed that GANT61-curcumin PLGA NPS significantly decreased cell viability and migration in MCF-7 cells by downregulating BMI1, PI3K, and GLI1 pathways [164].

### 5.7. Prostate Cancer

In prostate cancer, curcumin NPs have shown promising results in various experimental models. For example, Yan et al. demonstrated that docetaxel (DTX)-curcumin- Lipid Polymeric NPs (LPNs) significantly reduced cell viability and increased cytotoxicity in PC3 cells at a concentration of 3.62 ± 0.65 μM [165]. In vivo studies using PC3 cell-bearing mice treated with doses of 5 mg/kg and 10 mg/kg of DTX-Cur-LPNs revealed a decrease in tumor growth [165]. Similarly, curcumin-loaded cyclodextrin (CD)/cellulose nanocrystals (CNCx) NPs were tested on both PC3 and DU145 cell lines at concentrations ranging from 5–50 μM, where they effectively inhibited cell proliferation and reduced cell viability [131]. Another study by Adahoun et al. on PC3 cells treated with curcumin NPs (50–600 μM) also noted a decrease in cell viability and an increase in cytotoxicity [166].

Further advancements include the development of curcumin zinc liposomes, which effectively reduced cell viability in RM-1 cells at 0.8 ± 0.3 μg/mL, and curcumin copper liposomes, which had a similar effect at a concentration of 1.6 ± 0.3 μg/mL [66]. Moreover, Caldas et al. tested polyethyleneimine-encapsulated (PEC)-curcumin and PEC-tannic acid (PEC-T)-curcumin formulations on PC3 cells, where the PEC-curcumin variant exhibited a concentration-dependent decrease in cell viability at 1581 ± 96.3 μg/mL, while PEC-T-curcumin showed a greater effect at a lower concentration of 441.7 ± 52.0 μg/mL [167]. Lastly, curcumin-loaded peptide (Pep)-V1 and Pep-V2 nano-vesicles were tested on DU145 cells by Chen et al. [168]. The Pep-V1 formulation enhanced cytotoxicity at 7.0 ± 0.8 μM, while Pep-V2 achieved similar results at 13.3 ± 1.5 mM [168].

### 5.8. Brain Cancer

In the realm of glioma and neuroblastoma treatment, curcumin-loaded NPs have shown significant promise due to their enhanced bioavailability and improved therapeutic efficacy. Several studies have explored diverse curcumin formulations to target cancer cells, reduce cell viability, and promote apoptosis, highlighting curcumin’s versatility as an anticancer agent.

For example, Dey et al. investigated alginate-curcumin gold nanoparticles (MP@Alg–Ccm AuNPs) in C6 glioma cells [79]. At concentrations of 21 and 42 μM, these NPs significantly decreased cell viability and increased cytotoxicity, illustrating their potential in glioma treatment [79].

Kalashnikova et al. explored the effects of ceria NPs coated with curcumin on IMR-32 and SMS-KAN neuroblastoma cells [169]. At 100 μM, this formulation not only reduced cell viability but also induced apoptosis by modulating key molecular pathways, such as decreasing Bcl-2/Bax ratios, increasing caspase-3/7 activity, ROS, and HIF-1α [169].

Tian et al. developed hyaluronic acid (HA)-s-s-curcumin NPs, which were tested on G422 glioma cells. This formulation demonstrated a concentration-dependent reduction in cell viability (1–30 μg/mL) while significantly increasing cytotoxicity, further validating curcumin’s role in glioma treatment [170].

Another notable study by Hesari et al. used curcumin nano-micelles to target U-373 glioblastoma cells. These nano-micelles, at concentrations of 0.31–80 mg/mL, inhibited cell growth and invasion, induced tumor shrinkage, and triggered apoptosis by downregulating key signaling molecules like NF-κB, IκB, cyclin D1, survivin, axin, and E-cadherin [171].

Yadav et al. studied a carbon nitride nanohybrid loaded with curcumin in C6 glioma cells. At concentrations of 1–7.5 μM, this nanohybrid reduced cell viability and enhanced cytotoxicity by increasing ROS levels, leading to significant cell death [172].

He et al. investigated curcumin loaded in methoxy polyethylene glycol-polylactic acid (MPEG-PLA) and folate-conjugated polyethylene glycol-polylactic acid (Fa-PEG-PLA) NPs in GL261 glioma cells. With a concentration range of 0.3–25 μg/mL, this formulation induced apoptosis and decreased both cell growth and viability [173]. In GL261 cell-bearing mice, curcumin (50 mg/kg) led to reduced tumor growth and angiogenesis, further promoting apoptosis [173].

Pham et al. developed indocyanine green (ICG)/curcumin-loaded albumin NPs, which were tested in N2a neuroblastoma cells. These NPs decreased cell viability and increased cytotoxicity and apoptosis at concentrations of 0.3–40 μg/mL. In N2a xenograft-bearing mice, a curcumin equivalent of 5 mg/kg reduced tumor growth, demonstrating its potential in neuroblastoma therapy [174].

In the same vein, Zhang et al. assessed curcumin-loaded micelles in C6 glioma cells. With an IC_50_ 2.05 μg/mL, these micelles significantly decreased cell viability and increased cytotoxicity. Furthermore, it enhanced the intracellular release of curcumin, and increased the inhibition effect of cancer cells further establishing curcumin’s anticancer potential in glioma [175].

Sharma et al. tested curcumin carbon dots in C6 glioma cells, showing that concentrations between 31.25 and 500 μg/mL reduced cell viability, inhibited migration, and promoted apoptosis by increasing ROS and disrupting actin filaments and tubulin [176]. Hemmati et al. examined curcumin and chitosan-loaded nanocarriers in U87 MG glioblastoma cells. At concentrations of 5–30 μg/mL, this formulation led to a reduction in cell viability and an increase in apoptosis, highlighting its potential for glioblastoma treatment [177]. Gallien et al. encapsulated curcumin in dendrimers, which were tested in GL269, F98, and U87 glioma cells. This encapsulation, at curcumin equivalents of 0.02, 0.06, and 0.1 mg/mL, significantly decreased cell viability, making it a promising delivery system for curcumin in cancer treatment [178].

Chibh et al. focused on DOX-curcumin-loaded amino acid-based microbowls, targeting C6 glioma cells. At doses of 2.5–20 μL, these microbowls reduced cell viability and enhanced cytotoxicity by increasing ROS levels, providing a novel approach to enhance the efficacy of curcumin-based therapies [179]. Javed et al. explored curcumin and piperine-loaded lignin-grafted gold nanogels in U-251 MG glioblastoma cells. At concentrations of 0.6–1000 μM, the nanogels decreased cell viability and increased cytotoxicity and apoptosis by upregulating caspase-3 activity, demonstrating their potential as effective therapeutic agents [73].

Finally, Wanjale et al. evaluated curcumin-loaded polycaprolactone (PCL)-polyethylene glycol (PEG) co-polymers in U-251 glioblastoma cells. This formulation increased cytotoxicity and apoptosis at 1 mM in vitro. In U-251 cell-bearing mice, doses of 100–125 μL led to reduced tumor growth and decreased Ki-67 expression, further confirming curcumin’s anticancer potential [180].

Together, these studies demonstrate the vast therapeutic potential of curcumin-loaded NPs and formulations in targeting glioma and neuroblastoma, offering promising strategies for enhancing its efficacy and overcoming the limitations of conventional treatments.

### 5.9. Pancreatic Cancer

CNP systems have also shown considerable promise in targeting pancreatic cancer, one of the most aggressive and treatment-resistant malignancies.

Sivakumar et al. developed AS1411-conjugated curcumin-loaded superparamagnetic iron oxide nanoparticles (PLGA-SPIONS) and tested them on Panc1 and Mia-Pa-Ca-2 cells [181]. The results indicated a significant reduction in cell viability at concentrations ranging from 10 to 500 μg/mL, demonstrating the potential of this targeted delivery system for pancreatic cancer therapy [181]. Thakkar et al. explored curcumin-conjugated solid lipid NPs (curcumin c-SLNs) in LSL-Kras G12D/+; Pdx-1 Cre/+ transgenic mice, a model mimicking human pancreatic ductal adenocarcinoma (PDAC). Administering doses of 4.5, 45, and 135 mg/kg led to a notable decrease in tumor incidence, highlighting curcumin’s chemopreventive properties [182].

Another intriguing study by Madamsetty et al. investigated the combined effect of PEGylated irinotecan and curcumin-loaded nanodiamonds on AsPC-1 and Panc1 cells [183]. This combination significantly reduced cell viability and enhanced cytotoxicity at concentrations between 10 and 100 μg/mL. In KPC mice, the administration of 15 mg/kg curcumin equivalents not only suppressed tumor growth but also reduced Ki-67 expression and increased cleaved caspase-3, markers indicative of diminished cell proliferation and enhanced apoptosis [183].

Curcumin encapsulated in gelatin nanomaterials (Cur/gelatin NMs) was evaluated by Cheng et al. in multiple PDAC cell lines, including T3M4, Mia-Pa-Ca-2, and Panc1 cells [184]. The formulation led to reduced cell viability and increased cytotoxicity, apoptosis, and ER stress, evidenced by increased ROS, cleaved caspase-3, and the unfolded protein response markers Bip and p-PERK. In PDAC-bearing C57BL/6 mice, curcumin/gelatin NMs also inhibited tumor growth and proliferation while promoting ER stress, further reducing p-STAT3 expression and enhancing Bip expression [184].

In another study, Jadid et al. developed a nanoformulation combining hydroxytyrosol and curcumin (PLGA-PPA Hyd curcumin) and tested it on Panc1 cells [185]. This formulation significantly inhibited cell viability, colony formation, and migration, while inducing apoptosis, nuclear fragmentation, and cell shrinkage. Mechanistically, it was associated with reduced expression of MMP-2, MMP-9, and Bcl-2, along with increased Bax and caspase-9 levels, further validating its pro-apoptotic effects [185].

Finally, Zhou et al. evaluated the cytotoxic potential of curcumin encapsulated in zinc and copper liposomes in Panc1 cells. The curcumin-Zn ions liposomes exhibited an IC_50_ of 6.1 ± 1.0 μg/mL, while curcumin- metal liposomes demonstrated a stronger effect with an IC_50_ of 1.4 ± 0.4 μg/mL [66]. Both formulations effectively reduced cell viability and increased cytotoxicity, positioning liposomal curcumin as a potent option for pancreatic cancer treatment.

These studies collectively highlight the versatility of CNP systems in enhancing the therapeutic efficacy of curcumin in pancreatic cancer, offering promising avenues for future clinical applications.

### 5.10. Cervical Cancer

Curcumin NPs have shown promising potential in the treatment of cervical cancer by enhancing cytotoxicity, reducing cell viability, and inhibiting tumor growth in various in vitro and in vivo models. ZnFe_2_O_4_ curcumin NPs were tested on HeLa cells with a concentration range of 0.4–1 μg/mL, showing a decrease in cell viability and an increase in cytotoxicity [186].

PLGA nano-curcumin was tested on Caski and Siha cells, with concentrations ranging from 2.5 to 25 μM, resulting in decreased cell viability, increased apoptosis, G1/S arrest, and reduced oncogenic effects of BaP, migration, clonogenic potential, and cell proliferation. This was linked to the downregulation of miR-21, IL-6, p-STAT3, p-STAT5, NF-κB, p-PTEN, and nuclear translocation of β-catenin, while upregulating miR-214 and phosphatase and TENsin homolog (PTEN) [187]. In orthotopic Caski cells bearing NSG mice, treatment led to a decrease in tumor growth along with reductions in Ki-67, E6, E7, miR-21, and an increase in PTEN [187].

Curcumin-loaded halloysite nanotubes grafted with chitosan (HNTs-g-CS) NPs were also evaluated on Caski cells at concentrations greater than 64 μM, resulting in a decrease in cell viability [123]. Exosomal E-curcumin was assessed on HeLa and Caski cells at a concentration of 12.5 μM curcumin, leading to a decrease in cell survival [188], and in Caski cells bearing athymic nude mice, it reduced tumor growth with a dose of 20 mg/kg curcumin [188].

Curcumin-entrapped in PLGA-PEG nanoparticles conjugated to folic acid (PPF-curcumin) was tested on HeLa cells at 5 μM, showing reduced cell viability, increased cytotoxicity, and chemo sensitization, along with downregulation of NF-κB, p-Akt, p-p38, p-JNK, p-ERK1/2, COX-2, Bcl-2, cyclin D1, XIAP, c-IAP, and survivin [189]. A similar reduction in tumor growth was observed in HeLa cells bearing NOD-SCID mice at 25 mg/kg, with chemo sensitization and decreases in NF-κB, Cyclin D1, PCNA, and VEGF [189].

Further studies involving curcumin-loaded d-α-tocopherol (α-TOS)/lipid-based copolymeric nanomicellar system (VPM) (both transferrin-targeted and untargeted) on HeLa cells at 3–50 μg/mL led to reduced cell viability and tumor growth along with increased cytotoxicity [190]. The A- or S-polyactive curcumin system was tested on HeLa cells at 2.5 and 5 μg/mL, reducing cell viability, increasing cytotoxicity, and inducing apoptosis via an increase in ROS [191]. Similarly, the curcumin@ZIF-8/hyaluronic acid (HA) system on HeLa cells at 12.5–100 μg/mL decreased cell viability and increased cytotoxicity [192].

Other systems like curcumin-loaded silk nano discs, Fe_3_O_4_@PLGA-PEG@ folic acid FA, and SN_LYZ-BLA_-curcumin demonstrated similar efficacy in HeLa cells, with reduced cell growth and viability, or increased apoptosis and cytotoxicity [128,208,209]. In particular, ACPCSLNPs on HeLa cells (5–30 μM) resulted in reduced cell viability and increased cytotoxicity [193], while HES-curcumin NPs on HeLa cells (10–100 μg/mL) had a similar effect [141].

Curcumin-CBP, curcumin-LBP treatments led to decreased cell viability and increased cytotoxicity and apoptosis in HeLa cells (0.1–50 μg/mL), with in vivo results showing reduced tumor growth and apoptosis in HeLa cells bearing BALB/c mice at 2.5 mg/kg [194]. Additional nanocarriers like curcumin-loaded PEG oligodendron amphiphiles, curcumin-loaded C12-LBA nanovesicles, PEC-CUR, Poly@curcuminFA, and curcumin-conjugated YVO4 +/Yb3 + UC-MHNSPs also showed efficacy, reducing cell viability and inducing apoptosis [167,210,211,212].

### 5.11. Oral Cancer

In the context of oral cancer research, various formulations of curcumin NPs have demonstrated significant therapeutic potential. Srivastava et al. investigated nano curcumin in SCC090 cells, revealing a concentration-dependent reduction in cell viability (10–100 μg/mL), increased cytotoxicity, and enhanced chemotherapeutic effectiveness of 5-FU [195]. Notably, this treatment resulted in decreased ROS, increased Bax protein levels, and reduced Bcl-2 levels, suggesting a mechanism involving apoptosis induction.

Madeo et al. developed curcumin graphene oxide nanosheets blended into alginate hydrogels and tested these on SCC-25 cells. The results indicated a significant decrease in cell viability and increased cytotoxicity at concentrations of 2.5%, 5%, and 7.5% [196]. Fazli et al. explored curcumin-loaded niosomes in KB cells, with concentrations ranging from 4 to 32 μg/mL, which similarly resulted in decreased cell viability and increased cytotoxicity [213]. In an in vivo study, they also treated Sprague Dawley rats with 4NQO-induced oral cancer using either a dose of 4 mg/kg or a mouthwash formulation, leading to a reduction in precancerous changes and dysplasia [213]. Furthermore, curcumin NPs were evaluated in SCC4 cells by Essawy et al., where treatment with concentrations ranging from 25 to 250 μg/mL resulted in decreased cell viability, reduced cell migration, increased apoptosis, and enhanced antioxidant activity [197].

Collectively, these studies underscore the promising role of CNPs as effective agents in the treatment of oral cancer.

### 5.12. Bone Cancer

In the field of bone cancer research, several curcumin formulations have shown promising anti-cancer effects. Fatima et al. studied curcumin-loaded polymeric NPs (Cur-loaded PECs) in U2OS cells and observed a significant reduction in cell viability and an increase in cytotoxicity at a concentration of 50 μM curcumin [157].

Somu and Paul investigated a novel formulation known as SNLYZ-BLA-CUR in MG63 cells, reporting a dose-dependent decrease in cell viability and increased cytotoxicity at concentrations of 30 and 120 μg/mL [128].

Zhang et al. explored a composite material consisting of titanium dioxide (TiO_2_)/polydopamine (pDA)/β-Cyclodextrin (β-CD)/Curcumin, which demonstrated reduced MMPo, enhanced cytotoxicity, increased cell shrinkage, and elevated apoptosis rates in MG63 cells treated with curcumin equivalents ranging from 0.2 to 1.6 mg/mL [198]. Additionally, this study reported increased lactate dehydrogenase activity and ROS production. In an in vivo model, UMR-106 cell-bearing mice showed decreased tumor volume and increased apoptosis following treatment [198].

Further research involving a polycaprolactone (PCL)/curcumin/ polydopamine @Selenium (PDA@Se) formulation in MG63 cells resulted in diminished cell viability, reduced cell proliferation, compromised cell membrane integrity, and increased cytotoxicity, alongside enhanced F-actin condensation and ROS levels [199]; in vivo study, treatment of UMR-106 cell-bearing mice led to decreased tumor cells, increased tumor cell death, and improved wound healing [199].

These findings collectively underscore the potential of CNPs as effective therapeutic agents in the treatment of bone cancer.

### 5.13. Esophageal Cancer

In esophageal cancer research, various formulations of curcumin have demonstrated significant anti-cancer effects. Hosseini et al. investigated nano curcumin’s impact on KYSE-30 cells, revealing that it reduced cell viability and increased cytotoxicity at concentrations ranging from 0.23 to 60 mg/mL, while also downregulating cyclin D1, a key regulator of the cell cycle [200].

Xu et al. studied curcumin-loaded poly (lactic-co-glycolic acid) NPs (Cur-PPLGA-N) in ECa109 cells, finding that they significantly decreased cell viability and enhanced cytotoxicity and apoptosis at concentrations between 1.25 and 125 μg/mL [201].

Gao et al., developed a biomimetic nano-targeting drug delivery system. PEG-TE10@PLGA@DOX-curcumin nanoparticles (PMPNs) were prepared by co-loading DOX and curcumin into poly (lactic-co-glycolic acid) (PLGA) nanoparticles, coated with TE10 cancer cell membranes and distearoyl phosphatidylethanola-mine-polyethylene glycol (DSPE-PEG). They reported a reduction in cell viability and colony formation alongside increased cytotoxicity at concentrations from 0.31 to 10 μg/mL [202]. This treatment was associated with elevated levels of cytochrome c, Bax, and cleaved caspase-3, indicating an activation of apoptotic pathways [202]. Furthermore, Balb/c mouse model of TE10/DOX xenograft, showed that the administration of a dose of 5 mg/kg led to decreased tumor growth and volume, alongside increased survival rates, apoptosis, and necrosis [202]. Similar results were observed with the (PLGA)@curcumin+DOX formulation, reinforcing the effectiveness of curcumin-loaded NPs in enhancing the therapeutic potential of DOX [202].

These findings underscore the promising role of curcumin-based therapies in combating esophageal cancer.

### 5.14. Stomach Cancer

In the context of stomach cancer, various curcumin-based nanocomposites and NP formulations have shown remarkable anti-cancer properties. Dhivya et al. explored curcumin-loaded polymethyl methacrylate (PMMA)-polyethylene glycol/zinc oxide (PEG/ZnO) nanocomposites in AGS cells, revealing a concentration-dependent decrease in cell viability, alongside increased cytotoxicity, apoptosis, and S-phase cell cycle arrest at doses between 0.0001 to 1 μg/mL [203]. Similarly, curcumin encapsulated in PMMA-AA/ZnO NPs effectively reduced cell viability and heightened cytotoxicity in AGS cells within the same concentration range [203].

Wu et al. examined the effects of curcumin-loaded sodium caseinate and calcium phosphate nanocomposites (Cur@NaCas)@CaP in MGC-803 cells, reporting reduced cell viability and increased cytotoxicity, with enhanced antioxidant activity at concentrations between 1 to 5 μg/mL [204].

Alam et al. showed that curcumin-loaded PLGA NPs in AGS cells inhibited cell proliferation and promoted apoptosis in a dose-dependent manner at concentrations between 5 to 40 μM [205].

In a separate study, Song et al. used curcumin-loaded liposomes in HGC-27 cells, which significantly increased cytotoxicity at concentrations ranging from 0.78 to 12.5 μM [206].

These findings demonstrate the potential of curcumin-based nanoparticles and nanocomposites as promising therapeutic approaches for stomach cancer.

## 6. Clinical Studies of CNPs

Numerous clinical studies have explored the safety, pharmacokinetics, and therapeutic potential of curcumin, especially in treating cancer and other human disorders. Curcumin has demonstrated significant promise in clinical settings, with its ability to halt or even prevent the development of cancer cells. Many of these clinical trials have revealed that nanocurcumin is beneficial in treating various cancers [214].

In cancer treatment specifically, several clinical trials have provided promising results. One study using curcumin nanomicelle in bladder cancer patients during chemotherapy showed that a daily dose of 160 mg significantly increased clinical response rates, while being well tolerated by patients with no significant side effects [215].

Another trial involving prostate cancer patients administered nano-curcumin at 120 mg/day for 3 days before and during radiotherapy. The study reported a decrease in radiation-induced proctitis, demonstrating its protective role during cancer therapy without any serious adverse effects [216].

In a trial conducted on patients who had undergone thyroidectomy for thyroid cancer, nano-curcumin was administered at a dose of 160 mg/day for 10 days. The study showed a reduction in micronuclei in lymphocytes, indicating a potential protective role against radiation-induced genetic damage, with no adverse side effects reported, marking it as a safe therapeutic option [217].

Moreover, a study involving breast cancer patients treated with nano-curcumin at a dose of 80 mg/day for 2 weeks demonstrated a reduction in radiation-induced skin reactions and pain, further reinforcing the beneficial effects of CNPs in reducing cancer treatment side effects [218].

Considering that inflammation is a key hallmark of cancer [219], a clinical trial demonstrated that nano-curcumin is both safe and effective in multiple sclerosis patients by restoring the frequency and functionality of regulatory T cells [220]; which are implicated in inflammation, invasion, and metastasis in cancer [219].

Nevertheless, it is important to note that as of 30 November 2024, no studies were found on the ClinicalTrials.gov website using “curcumin”, “nanoparticles”, “cancer”, and “tumor” as key words.

## 7. Challenges in the Development and Clinical Application of Curcumin-Loaded Nanoparticles

### 7.1. Toxicity Concerns

One of the significant limitations in using nanoparticles, especially those based on metallic materials (e.g., gold, silver, or iron oxide nanoparticles), is their potential for toxicity [221]. Metal-based nanoparticles can induce oxidative stress, inflammation, or even cytotoxic effects, particularly when they accumulate in vital organs such as the liver, spleen, and kidneys [221,222]. Additionally, the surface coatings, stabilizers, or by-products of nanoparticle degradation can contribute to adverse biological interactions [223]. Addressing these concerns requires comprehensive toxicological evaluations during preclinical testing, as well as the incorporation of biocompatible and biodegradable materials to mitigate harmful effects.

### 7.2. Stability Issues

The stability of curcumin-loaded nanoparticles under physiological conditions is critical for their therapeutic efficacy [224]. Curcumin, being inherently hydrophobic and chemically unstable, faces degradation challenges when exposed to light, heat, or alkaline environments [224]. In nanoparticle formulations, these challenges are compounded by issues such as particle aggregation, premature drug release, or loss of encapsulated curcumin during storage and administration [225,226]. Strategies to enhance stability include optimizing the nanoparticle composition with stabilizing agents, employing surface functionalization techniques, and incorporating polymers or lipid-based carriers that protect curcumin from environmental stressors.

### 7.3. Manufacturing and Scalability

Transitioning curcumin-loaded nanoparticles from laboratory research to large-scale clinical application involves significant hurdles in manufacturing. These challenges include achieving uniform particle size distribution, ensuring batch-to-batch reproducibility, and maintaining high encapsulation efficiency during production [227,228]. Moreover, the cost of raw materials, sophisticated equipment, and quality control measures can be prohibitive, particularly in resource-limited settings [229]. Scaling up production also necessitates stringent regulatory compliance to meet clinical standards for purity, safety, and efficacy. Innovative production techniques, such as microfluidics or high-shear mixing, may offer solutions to these scalability challenges.

### 7.4. Biodistribution and Clearance

The therapeutic success of curcumin-loaded nanoparticles depends on their ability to reach the target site effectively while minimizing off-target effects. However, the biodistribution of nanoparticles can be influenced by several factors, including their size, shape, surface charge, and coating materials [230]. For instance, smaller nanoparticles may penetrate deeper into tissues but face rapid clearance by the kidneys, whereas larger particles may be sequestered by the mononuclear phagocyte system (MPS) in the liver and spleen [231]. Addressing these issues requires designing nanoparticles with stealth properties, such as polyethylene glycol (PEG) coating, to evade immune detection and fine-tuning their size and surface characteristics to optimize biodistribution and clearance kinetics [232].

## 8. Conclusions

The therapeutic potential of curcumin, particularly when enhanced through NP-based delivery systems, offers a promising approach to tackling cancer and other complex diseases. This review has highlighted the advances in CNPs, which successfully address the compound’s natural limitations, such as low bioavailability, rapid metabolism, and poor solubility. CNPs, through polymeric, lipid-based, and inorganic NP formulations, enable targeted and sustained curcumin delivery, thereby amplifying its anticancer efficacy across multiple tumor types while maintaining a favorable safety profile. Clinical studies have demonstrated encouraging outcomes with nano-curcumin, improving treatment tolerability and reducing adverse effects associated with conventional cancer therapies. The versatility of CNPs extends beyond oncology, as seen in preliminary applications in autoimmune disorders and other chronic inflammatory diseases, positioning curcumin NPs as a multifaceted tool in modern medicine.

The future of curcumin-based nanotherapies holds considerable potential, yet several challenges remain in translating preclinical success into widespread clinical application. First, standardizing CNP formulations to ensure consistent pharmacokinetics and biodistribution is essential to enable clinical adoption. Additionally, developing efficient manufacturing processes that support large-scale production will be vital to meeting clinical and commercial demands. Further research should focus on exploring combinational therapies that utilize CNPs alongside existing chemotherapeutic agents, radiotherapy, and immunotherapy. Such synergistic approaches may enhance curcumin’s effectiveness against MDR cancers and enable personalized treatment regimens. Expanding clinical trials across diverse populations and cancer types will be critical to validating the broad applicability of CNPs and understanding potential patient-specific responses. Finally, innovations in nanotechnology, including dual-targeting NPs and responsive drug-release mechanisms, could further refine CNP delivery, optimizing therapeutic outcomes. With continued interdisciplinary efforts, curcumin-based nanotherapies are well-positioned to become a key component in the treatment arsenal against cancer and beyond.

## Figures and Tables

**Figure 1 pharmaceutics-17-00114-f001:**
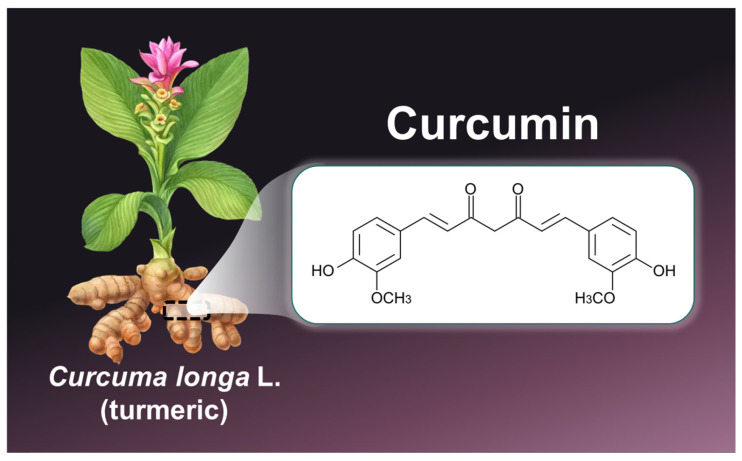
*Curcuma longa* L. and the chemical structure of curcumin, one of its main bioactive compounds.

**Figure 2 pharmaceutics-17-00114-f002:**
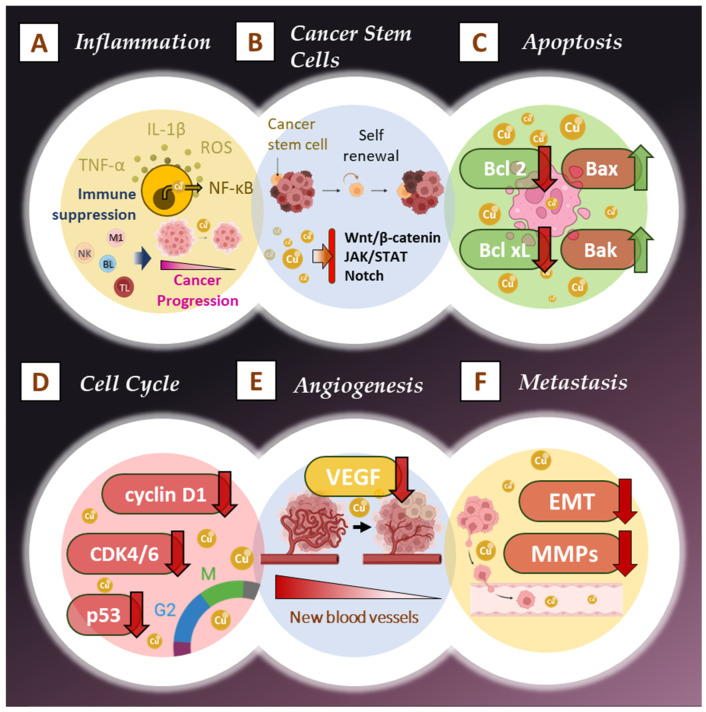
Curcumin anticancer activities; (**A**) Inflammation: curcumin inhibits inflammation by reducing the production of pro-inflammatory cytokines (e.g., IL-1β, TNF-α) and reactive oxygen species (ROS), which in turn suppresses NF-κB signaling and supports immune activity, potentially preventing cancer progression; (**B**) Cancer stem cells: curcumin hinders the self-renewal and maintenance of cancer stem cells by targeting key pathways such as Wnt/β-catenin, JAK/STAT, and Notch; (**C**) Apoptosis: curcumin promotes apoptosis by downregulating anti-apoptotic proteins (Bcl-2, Bcl-xL) and upregulating pro-apoptotic proteins (Bax, Bak), thus facilitating cancer cell death; (**D**) Cell cycle: curcumin disrupts cell cycle progression by modulating critical regulators like cyclin D1, CDK4/6, and p53, inhibiting cancer cell proliferation; (**E**) Angiogenesis: curcumin impedes angiogenesis by reducing vascular endothelial growth factor (VEGF) levels, which restricts the formation of new blood vessels that would otherwise nourish the tumor; (**F**) Metastasis: curcumin reduces metastasis by inhibiting epithelial-mesenchymal transition (EMT) and matrix metalloproteinases (MMPs), preventing cancer cells from invading and spreading to distant sites.

**Figure 3 pharmaceutics-17-00114-f003:**
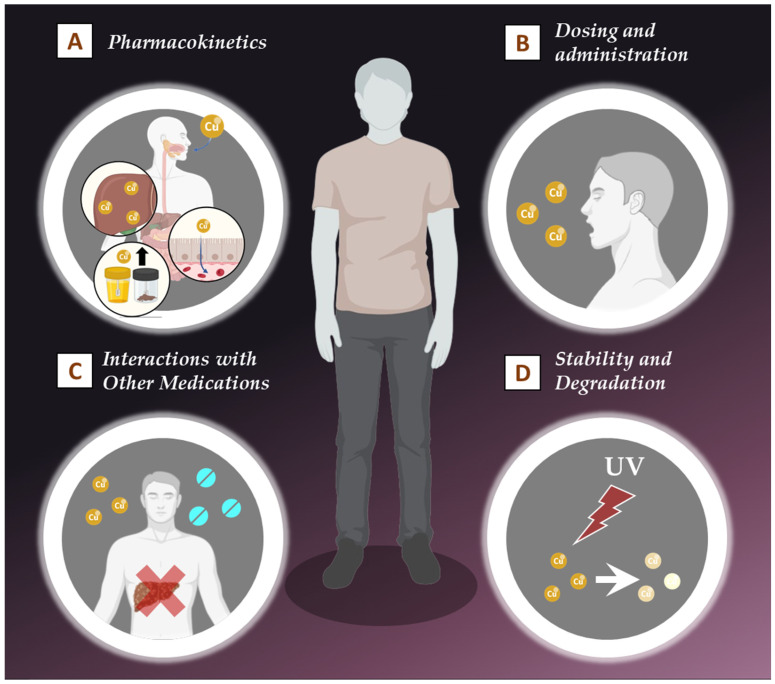
Challenges of curcumin in cancer therapy; (**A**) Curcumin pharmacokinetics, (**B**) Dosing of curcumin and its administration, (**C**) Curcumin interaction with other medications, (**D**) Curcumin stability and degradation.

**Figure 4 pharmaceutics-17-00114-f004:**
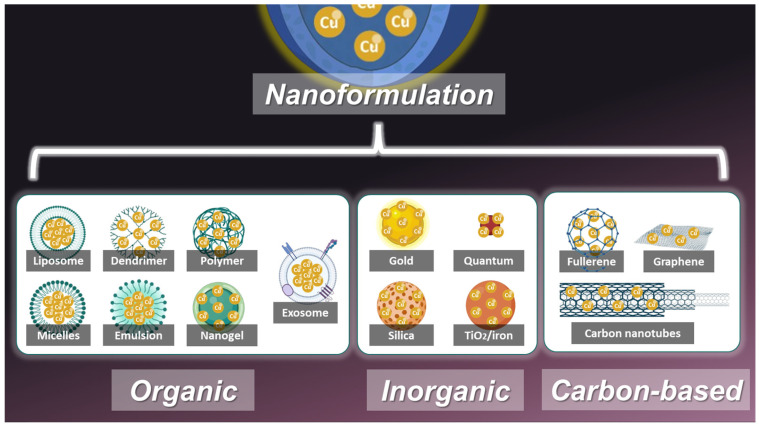
Classification of curcumin nanoformulations.

**Figure 5 pharmaceutics-17-00114-f005:**
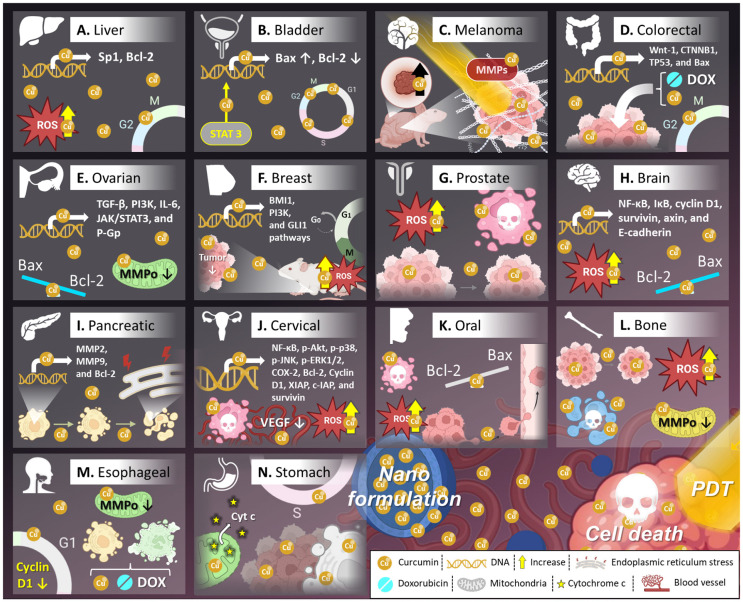
Effects of curcumin on various cancer types and mechanisms of tumor progression and cell death. (**A**) Liver cancer: curcumin-related upregulation of Sp1 and Bcl-2, increased ROS production, and cell cycle progression through G2 phase, (**B**) Bladder cancer: curcumin promotes STAT3 activation, upregulation of Bax, down regulation of Bcl-2, and cell cycle progression through G2/S phase, (**C**) Melanoma: curcumin-induced MMPs affecting tumor invasiveness and metastasis, (**D**) Colorectal cancer: curcumin influences Wnt-1, CTNNB1, TP53, and Bax expression, contributing to resistance to DOX and progression through the cell cycle, (**E**) Ovarian cancer: curcumin enhances TGF-β, PI3K, IL-6, JAK/STAT3, P-gp expression, and reduces Bcl-2, facilitating tumor progression, (**F**) Breast cancer: curcumin activates BM1, PI3K, and GLI1 pathways, promoting ROS generation and cancer cell survival, (**G**) Prostate cancer: curcumin induce ROS production and cancer cell death, leading to tumor growth inhibition, (**H**) Brain cancer: curcumin affects NF-κB, IκB, cyclin D1, survivin, axin, and E-cadherin, promoting ROS production and down regulation ofBcl-2 expression, contributing to cancer cell apoptosis, (**I**) Pancreatic cancer: curcumin induces MMP-2, MMP-9, and Bcl-2, associated with cancer cell invasiveness, (**J**) Cervical cancer: curcumin induces VEGF, ROS production, and activates survival pathways, including NF-κB, p-Akt, p38, p-JNK, p-ERK1/2, COX-2, Bcl-2, XIAP, c-IAP, and survivin, (**K**) Oral cancer: curcumin induces increased ROS production and downregulation of Bcl-2 expression, contributes to apoptosis and the inhibition of cancer progression, (**L**) Bone cancer: elevated ROS production and decreased MMPs with curcumin accumulation promote cell death., (**M**) Esophageal cancer: curcumin causes MMP activation, cyclin D1 downregulation, and resistance to DOX treatment, (**N**) Stomach cancer: curcumin triggers cytochrome c (Cyt c) release, initiating cell death.

**Figure 6 pharmaceutics-17-00114-f006:**
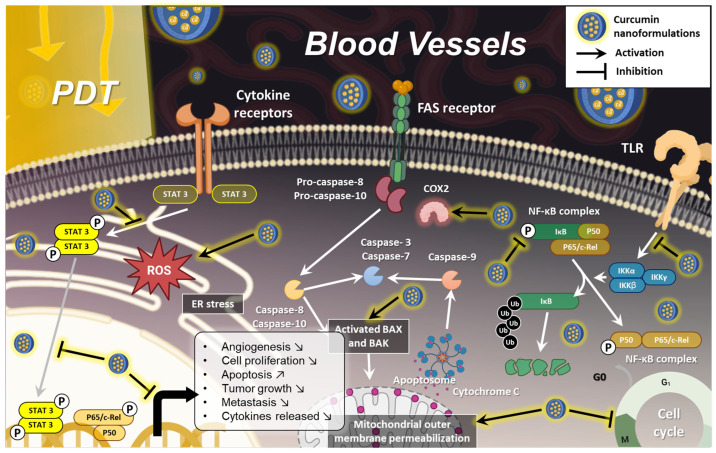
Anticancer activity of curcumin nanoformulations: mechanisms of action. Photodynamic therapy (PDT): curcumin nanoformulations, when activated by PDT, enhance the generation of ROS, leading to oxidative stress and ER stress, both of which contribute to cellular apoptosis. STAT3 pathway inhibition: curcumin suppresses the phosphorylation of STAT3, blocking its downstream effects, which include angiogenesis, cell proliferation, and tumor growth. This inhibition also contributes to a reduction in metastasis and cytokine release. FAS and caspase pathway activation: curcumin nanoformulations activate the FAS receptor, which triggers a cascade involving caspase-8, -9, and -3, leading to mitochondrial membrane permeabilization through BAX and BAK proteins. The release of cytochrome c into the cytoplasm then promotes apoptosome formation and drives cancer cell apoptosis. NF-κB pathway suppression: curcumin nanoformulations inhibit the NF-κB pathway by blocking the phosphorylation of IκB by IKK complex, thus preventing NF-κB’s translocation to the nucleus. This inhibition reduces pro-inflammatory cytokine release, which is associated with tumorigenesis and cancer progression. COX-2 inhibition: curcumin also targets COX-2, reducing inflammation and potentially impairing tumor survival and proliferation.

**Table 1 pharmaceutics-17-00114-t001:** Summary of preclinical studies related to the anticancer activity of CNPs in various types of cancers.

Liver Cancer				
Nanoparticles	Dosage	Cell Lines	Observed Effects	References
Mucoadhesive curcumin nanotherapy	2.5–20 μM	HuH7, HepG2	cell viability ↓, apoptosis ↑, necrosis ↑, cell proliferation ↓	[117]
Targeted dendrimeric curcumin	5–50 mM 0.5 mg/25 g	HuH7, Hepa1-6 Hepa1-6 xenograft models	cell viability ↓, G2/M phase cell cycle arrest, ROS levels ↑, depletion of ATP and glutathione ↓, tumor growth ↓, survival rates ↑	[67]
Curcumin-loaded nanoechinus	10 μg/mL	HepG2, HepG2-bearing mice	cytotoxicity ↑, tumor volume ↓, tumor weight ↓, cellular damage ↑	[118]
curcumin carbon nanodots	0.1–3.2 mg/mL	HuH7 and HepG2	apoptosis ↑, cell viability ↓	[119]
Fe_3_O_4_@curcumin-LDH/PDA	6.125–200 μg/mL	HepG2 cells and HUVECs	cytotoxicity ↑, cell viability ↓, ferroptosis ↑	[120]
KGM-gAH8 micelles	2–266 μg/mL	HepG2 cells	cytotoxicity ↑, cell viability ↓	[86]
cisplatin-curcumin coloaded liposomes + cisplatin	9.8 mg/kg	HepG2 and H22 xenografts	Sp1 ↓, Bcl-2 ↓, ROS ↑, p53 ↑, caspase-3 ↑, Bax ↑	[121]
Bladder cancer				
Nanoparticles	Dosage	Cell lines	Observed Effects	References
Dendrosomes	17.5 µM	5637	bioavailability ↑, apoptosis ↑, cell cycle arrest ↑, stemness markers (Oct4, SOX2, NANOG) ↓	[122]
HNTsg-CS NPs	5.3 µM	EJ-1	cell viability ↓, apoptosis ↑, cell cycle arrest ↑, controlled release ↑, curcumin stability ↑	[123]
Copper NPs	290 µg/mL	TCCSUP	apoptosis ↑, Bax ↑, cleaved caspase-8 ↑, Bcl-2 ↓, STAT3 pathway ↓, colony formation ↓	[124]
Melanoma				
Nanoparticles	Dosage	Cell lines	Observed Effects	References
curcumin and chrysin-loaded NPs	5–60 μM 30 mg/kg	B16F10, B16F10-bearing C57BL/6 mice	MMP-2 ↓, MMP-9↓, TERT↓, TIMP-1↑, TIMP-2↑, tumor growth↓	[125]
Au-Lipos Cur NP	200 μg/mL	B16 cells B16 bearing mice	cell growth ↓, viability ↓, PTT sensitivity ↑, Hsp70 ↓, SLUG ↓, Mucin ↓, tumor growth ↓	[126]
PPCP nanofibrous matrix	Not specified	A375 cells A375 bearing mice	tumor growth ↓, cell viability↓, apoptosis ↑, Ki-67 ↓	[127]
SN_LYZ-BLA_-curcumin	40–160 µg/mL	B16F10 cells	cell viability ↓, cytotoxicity ↑	[128]
Zn and Cu liposomes	4.3 µM	B16F10 cells	cell viability ↓, proliferation ↓	[66]
Colorectal Cancer				
Nanoparticles	Dosage	Cell lines	Observed Effects	References
PEGylated curcumin NPs	1–40 μM 10 mg/kg	CT26 cells CT26-bearing nude mice	cell viability ↓, cytotoxicity ↑, tumor volume ↓, tumor weight ↓	[129]
cyclodextrin/carboxymethylcellulose NPs	0.01–100 μg/mL	HCT116 cells	cytotoxicity ↑, G2/M cell cycle arrest, apoptosis ↑	[130]
Cyclodextrin/carboxymethylcellulose NPs	5–50 mM	HT29 cells	cell proliferation↓, viability ↓	[131]
Phyto/active gold-fluorescein/chitosan biohybrid	2.5%–35%	HT29 cells	cell viability ↓, cytotoxicity ↑	[132]
Curcumin encapsulated in hydroxypropyl-β-cyclodextrin	20–60 µM	COLO205 cells	DOX sensitivity ↑, cell viability ↓	[133]
Liposomes co-loaded with curcumin and doxorubicin	0.45–41.85 mM	C26 cells	cell proliferation ↑	[134]
Curcumin + 5-fluorouracil (5-FU) delivery system	1.5–25 μg/mL	HCT116 cells	cell viability ↓	[135]
Curcumin-loaded PLGA/PEG NPs	12.05 μM (Caco-2)	Caco-2 cells	proliferation ↓, cytotoxicity ↑, hTERT expression ↓	[136]
Curcumin-loaded PLGA/PEG NPs	3.5–60 μM	SW480 cells	cell proliferation ↓, cytotoxicity ↑	[137]
Liposomal curcumin + DOX	20 μM	C26 cells	proliferation ↓, cytotoxicity ↑, cytokines and growth factors ↓	[138]
Curcumin-loaded nanocomposite	62.5–1000 μg/mL	HT29, SW948 cells	cytotoxicity ↑	[139]
Curcumin-loaded protamine NPs	2.8–4.5 μg/mL	SW480 cells	cell viability ↓, migration ↓, miR-145 ↑, IGF-1R ↓	[140]
Mesoporous silica NPs loaded with curcumin	10–200 μg/mL	LS174T cells	cell viability ↓, cytotoxicity ↑	[80]
Curcumin-loaded NPs	10–100 μg/mL	Caco-2 cells	viability ↓, cytotoxicity ↑	[141]
Curcumin-loaded NPs	50 μM	HCT116 cells	viability ↓, apoptosis ↑	[142]
Solid iron-curcumin NPs	0.028–3.6 mg/mL	HT-29 cells	cell viability ↓, tumor volume ↓	[143]
Cur-Res-BS	100 to 1000 μg/mL	HCT116, Caco-2	cell viability↓, anti-proliferative and pro-apoptotic effects ↑, modulation of cancer-related genes (Wnt-1, CTNNB1, TP53, Bax)	[144]
Ovarian cancer				
Nanoparticles	Dosage	Cell lines	Observed Effects	References
Curcumin niosomes	0.05–0.2 μg/mL 2.5 mg/kg	A2780s, A2780cp-1 BALB/c mice	cytotoxicity ↑, apoptosis ↑, NF-κB ↓, p53 ↑, tumor size ↓	[145]
Gemini-curcumin formulation	100 µM	OVCAR-3	cell proliferation ↓, apoptosis ↑, modulation of Bax/Bcl-2	[146]
Curcumin-loaded Fe_3_O_4_ NPs	0.01 mg/mL	SKOV-3	cell viability ↓, cancer cell growth ↓	[147]
F68-Cis–Cur formulation	0.9375–30 μM	A2780	cell viability ↓, apoptosis ↑, MMPo ↓	[148]
Curcuminδ-T3 nanoemulsion	25 μM	OVCAR-8	cell viability ↓, apoptosis ↑, caspase activation ↑, NF-κB ↓	[149]
Curcumin-loaded NPs	10 μM	A2780, A2780/ADM	multidrug resistance ↓, P-gp expression ↓	[150]
PEG-PDLLA NPs	1 mg/mL	A2780	cell proliferation ↓, cell migration ↓, modulation of NF-κB/PRL-3 signaling pathway	[151]
Curcumin NPs + Cisplatin	20 and 40 μM	DMBA-induced ovarian cancer rat model	tumor volume ↓, increased apoptosis ↑, modulation of TGF-β, PI3K, IL-6, and JAK/STAT3 pathways	[152]
Breast cancer				
Nanoparticles	Dosage	Cell lines	Observed Effects	References
Dendrosomal curcumin	40 and 80 mg/kg	4T1 mice	tumor incidence ↑ tumor volume↑, IL-10 ↓, STAT3 ↓, arginase I ↓, IL-12 ↑, STAT4 ↑	[153]
Polymeric micelles (curcumin + DOX)	0.01–1 mg/mL	MCF-7, MCF-7/ADR, 4T1	cell viability ↓, cytotoxicity apoptosis ↑, tumor growth ↓, tumor volume ↓	[154]
Gum arabic-curcumin micelles	25–0.78 g/mL	MCF-7	cell viability ↓, cytotoxicity ↑	[155]
MP@Alg–curcumin AuNPs	42 µM	MCF-7	cell viability ↓, cytotoxicity ↑	[79]
Curcumin-P123-PAE	20–100 µg/mL	MCF-7	cell viability ↓, cytotoxicity ↑	[156]
HNTs-g-CS NPs (curcumin loaded)	5.3–192 μM	MCF-7	cell viability ↓	[123]
PECs (curcumin loaded)	50 µg/mL	MDA-MB-231	cytotoxicity ↑, apoptosis ↑, cell cycle arrest at G0/G1 phase	[157]
GP-Cur-Ptx	1.450 μg/mL	MDA-MB-231	apoptosis ↑, cytotoxicity ↓, ROS production ↑	[125]
Chitosan/perfluorohexane nanodroplets	0.004–0.4 μg/mL	4T1	cell viability ↓, cytotoxicity ↑	[158]
FPCHN-30 (folate conjugated lipid NPs)	5 nM	MCF-7/ADR	cytotoxicity ↑, P-gp expression ↓	[159]
Metformin-curcumin-PLGA/PEG NPs	50 µM	T47D	cytotoxicity ↑, hTERT ↓	[160]
Curcumin encapsulated NPs	15 µM	SK-BR-3	apoptosis ↑, Bcl-2 ↓, MMP-9 ↓	[161]
ALN-oHA-S-S-curcumin	1.25–40 μg/mL	MCF-7, MDA-MB-231	cytotoxicity ↑	[162]
Icariin and curcumin-loaded micelles	10 μg/mL (200 μL) 0.5 mg/mL	MCF-7 MCF-7 bearing mice	invasion ↓, tumor growth ↓	[163]
GANT61-curcumin PLGA NPs	0.1–1 mg/mL	MCF-7	cell viability↓, migration ↓, BMI1, PI3K, and GLI1 pathways ↓	[164]
Prostate cancer				
Nanoparticles	Dosage	Cell lines	Observed Effects	References
DTX-Cur-Lipid Polymeric NPs (LPNs)	3.62 mΜ 5–10 mg/kg	PC3 PC3-bearing mice	cell viability↓, increased cytotoxicity ↑, decreased tumor growth ↓	[165]
Curcumin-loaded CD/CNCx NPs	5–50 μM	PC3, DU145	cell proliferation ↓, cell viability ↓	[131]
Curcumin NPs	50–600 μM	PC3	cell viability↓, cytotoxicity ↑	[166]
Curcumin zinc liposomes	0.8 μg/mL	RM-1	cell viability↓	[66]
Curcumin copper liposomes	1.6 μg/mL	RM-1	cell viability↓	[66]
PEC-Curcumin	1581 μg/mL	PC3	cell viability↓	[167]
PEC-Tannic acid-Curcumin	441.7 μg/mL	PC3	cell viability↓	[167]
Curcumin-loaded Pep-V1 nano-vesicles	7 μM	DU145	cytotoxicity ↑	[168]
Curcumin-loaded Pep-V2 nano-vesicles	13.3 mM	DU145	cytotoxicity ↑	[168]
Brain cancer				
Nanoparticles	Dosage	Cell lines	Observed Effects	References
Alginate-curcumin gold nanoparticles (MP@Alg–Ccm AuNPs)	21 and 42 μM	C6 glioma cells	cell viability ↓, cytotoxicity ↑	[79]
Ceria NPs coated with curcumin	100 μM	IMR-32 and SMS-KAN neuroblastoma cells	cell viability ↓, apoptosis ↑, modulation of Bcl-2/Bax ratio, caspase-3/7 ↑, ROS ↑, HIF-1α ↑	[169]
Hyaluronic acid (HA)-s-s-curcumin NPs	1–30 μg/mL	G422 glioma cells	cell viability ↓, cytotoxicity ↑	[170]
Curcumin nano-micelles	0.31–80 mg/mL	U-373 glioblastoma cells	cell growth ↓ and invasion ↓, apoptosis ↑, NF-κB ↓, IκB ↓, cyclin D1 ↓, surviving ↓, axin ↓, E-cadherin ↓	[171]
Carbon nitride nanohybrid loaded with curcumin	1–7.5 μM	C6 glioma cells	cell viability ↓, increased cytotoxicity ↑, ROS-mediated pathways ↑	[172]
MPEG-PLA and Fa-PEG-PLA curcumin NPs	0.3–25 μg/mL	GL261 glioma cells	apoptosis ↓, cell growth ↓, cell viability ↓; tumor growth ↓, angiogenesis ↓	[173]
ICG/curcumin-loaded albumin NPs	0.3–40 μg/mL 5 mg/kg	N2a neuroblastoma cells N2a xenograft-bearing mice	cell viability ↓, cytotoxicity ↓ and apoptosis ↑, tumor growth ↓	[174]
Curcumin-loaded micelles	2.05 μg/mL	C6 glioma cells	cell viability ↓, intracellular curcumin release ↑, proliferation ↓	[175]
Curcumin carbon dots	31.25–500 μg/mL	C6 glioma cells	cell viability ↓, inhibited migration, promoted apoptosis via ROS increase, disrupted actin filaments and tubulin	[176]
Curcumin and chitosan-loaded nanocarriers	5–30 μg/mL	U87 MG glioblastoma cells	cell viability ↓, apoptosis ↑	[177]
Curcumin encapsulated in dendrimers	0.02–0.1 mg/mL	GL269, F98, and U87 glioma cells	cell viability ↓	[178]
DOX-curcumin-loaded amino acid-based microbowls	2.5–20 μL	C6 glioma cells	cell viability ↓, cytotoxicity ↑, ROS-mediated mechanisms ↑	[179]
Curcumin and piperine-loaded lignin-grafted gold nanogels	0.6–1000 μM	U-251 MG glioblastoma cells	cell viability ↓, cytotoxicity ↑, apoptosis ↑, caspase-3 activity ↑	[73]
Curcumin-loaded PCL-PEG co-polymers	1 mM, 100–125 μL	U-251 glioblastoma cells	cytotoxicity ↑, apoptosis ↑, tumor growth ↓, Ki-67 expression ↑	[180]
Pancreatic cancer				
Nanoparticles	Dosage	Cell lines	Observed Effects	References
AS1411-conjugated curcumin-loaded superparamagnetic iron oxide nanoparticles (PLGA-SPIONS)	10–500 μg/mL	Panc1, Mia-Pa-Ca-2	cell viability ↓	[181]
Curcumin-conjugated solid lipid nanoparticles (curcumin c-SLNs)	4.5–135 mg/kg	LSL-Kras G12D/+; Pdx-1 Cre/+ (PDAC model)	tumor incidence ↓	[182]
PEGylated irinotecan and curcumin-loaded nanodiamonds	10–100 μg/mL; 15 mg/kg	AsPC-1, Panc1, KPC mice	cell viability ↓, cytotoxicity ↑, tumor growth ↓, Ki-67 expression ↓, cleaved caspase-3 ↑	[183]
Curcumin encapsulated in gelatin nanomaterials (Cur/gelatin NMs)	0.6–2 mg/mL	T3M4, Mia-Pa-Ca-2, Panc1	cell viability ↓, cytotoxicity ↑, apoptosis ↑, ER stress ↑, tumor growth ↓, p-STAT3 ↓, Bip expression ↓	[184]
PLGA-PPA hydroxytyrosol and curcumin nanoformulation	10–320 μM	Panc1	cell viability ↓, colony formation ↓, migration ↓; apoptosis ↑, nuclear fragmentation ↑, cell shrinkage↑, reduced MMP-2 ↓, MMP-9 ↓, Bcl-2 ↓, Bax ↑, caspase-9 ↑	[185]
Curcumin encapsulated in zinc and copper liposomes	6.1 μg/mL	Panc1	cell viability ↓, cytotoxicity ↑	[66]
Cervical cancer				
Nanoparticles	Dosage	Cell lines	Observed Effects	References
ZnFe_2_O_4_ curcumin NPs	0.4–1 μg/mL	HeLa	cell viability ↓, cytotoxicity ↑	[186]
PLGA nano-curcumin	2.5–25 μM	Caski, Siha Caski (orthotopic, NSG mice)	cell viability ↓, apoptosis ↑, G1/S arrest, BaP oncogenic effects ↓, migration ↓, clonogenic ↓, cell proliferation ↓, miR-21 ↓, IL-6 ↓, p-STAT3 ↓, p-STAT5 ↓, NF-κB ↓, p-PTEN ↓, nuclear β-catenin ↓, miR-214 ↑, PTEN ↑, tumor growth ↓, decreased Ki-67 ↓, E6 ↓, E7 ↓, miR-21 ↓, increased PTEN ↑	[187]
Curcumin-loaded HNTs-g-CS NPs	>64 μM	Caski	cell viability ↓	[188]
Exosomal E-curcumin	12.5 μM 20 mg/kg	HeLa, Caski	tumor growth ↓	[188]
PLGA-PEG-curcumin conjugated to folic acid	5 μM 25 mg/kg	HeLa HeLa (NOD-SCID mice)	cell viability ↓, cytotoxicity ↑, chemo sensitization ↑, NF-κB ↓, p-Akt ↓, p-p38 ↓, p-JNK ↓, p-ERK1/2 ↓, COX-2 ↓, Bcl-2 ↓, cyclin D1 ↓, XIAP ↓, c-IAP ↓, surviving ↓, tumor growth ↓, chemo sensitization ↑, NF-κB ↓, Cyclin D1 ↓, PCNA ↓, VEGF ↓	[189]
α-TOS/lipid-based copolymeric nanomicellar VPM	3–50 μg/mL	HeLa	cell viability ↓, tumor growth ↓, cytotoxicity ↑	[190]
A- or S-polyactive curcumin system	2.5–5 μg/mL	HeLa	cell viability ↓, increased cytotoxicity ↑, apoptosis ↑, ROS ↑	[191]
Curcumin@ZIF-8/hyaluronic acid	12.5–100 μg/mL	HeLa	cell viability ↓, cytotoxicity ↑	[192]
ACPCSLNPs	5–30 μM	HeLa	cell viability ↓, cytotoxicity ↑	[193]
HES-curcumin NPs	10–100 μg/mL	HeLa	cell viability ↓, cytotoxicity ↑	[141]
Curcumin-CBP, Curcumin-LBP	0.1–50 μg/mL 2.5 mg/kg	HeLa (BALB/c mice)	cell viability ↓, cytotoxicity ↑, apoptosis↑, tumor growth ↓, apoptosis ↑	[194]
Oral cancer				
Nanoparticles	Dosage	Cell lines	Observed Effects	References
Nano curcumin	10–100 μg/mL	SCC090	Reduced cell viability, cytotoxicity ↑, enhanced chemotherapeutic effectiveness ↑, Bax ↑, Bcl-2 ↓, apoptosis ↑	[195]
Curcumin graphene oxide nanosheets in alginate hydrogel	2.5%, 5%, and 7.5%	SCC-25	cell viability ↓, cytotoxicity↑	[196]
Curcumin-loaded niosomes	4–32 μg/mL 4 mg/kg	KB Sprague Dawley rats 4NQO-induced oral cancer	cell viability↓, cytotoxicity ↑, precancerous changes ↓, dysplasia↓	[143]
Curcumin nanoparticles	25–250 μg/mL	SCC4	cell viability ↓, migration ↓, apoptosis ↑, antioxidant activity ↑	[197]
Bone cancer				
Nanoparticles	Dosage	Cell lines	Observed Effects	References
Curcumin-loaded PECs	50 μM	U2OS	cell viability ↓, cytotoxicity ↑	[157]
SNLYZ-BLA-CUR	30 and 120 μg/mL	MG63	cell viability↓, cytotoxicity ↑	[128]
TiO2/pDA/β-CD/Curcumin	0.2 to 1.6 mg/mL	MG63	MMPo ↓, cytotoxicity ↑, cell shrinkage ↑, apoptosis ↑, lactate dehydrogenase activity ↑, ROS production ↑	[198]
PCL/curcumin/pDA@Se	nd *	MG63 UMR-106 cell-bearing mice	cell viability ↓, cell proliferation ↓, cell membrane integrity ↓, cytotoxicity ↑, F-actin condensation ↑, ROS levels↑, tumor cells ↓, tumor cell death ↓, wound healing ↑	[199]
Esophageal cancer				
Nanoparticles	Dosage	Cell lines	Observed Effects	References
Nano curcumin	0.23 to 60 mg/mL	KYSE-30	cell viability ↓, cytotoxicity ↑, cyclin D1 ↓	[200]
Curcumin-loaded PLGA NPs (Cur-PPLGA-N)	1.25 to 125 μg/mL	ECa109	cell viability ↓, enhanced cytotoxicity and apoptosis	[201]
PEG-TE10@PLGA@DOX-curcumin nanoparticles (PMPNs) (PLGA)@curcumin + DOX	0.31 to 10 μg/mL 5 mg/kg	TE10 TE10/DOX xenograft model	cell viability ↓ and colony formation ↓, cytotoxicity ↑, cytochrome c ↑, Bax ↑, cleaved caspase-3 levels ↑, tumor growth ↓, tumor volume ↓, survival rates ↑, apoptosis ↑, necrosis ↑	[202]
Stomach cancer				
Nanoparticles	Dosage	Cell lines	Observed Effects	References
Curcumin-loaded PMMA-PEG/ZnO nanocomposites	0.0001–1 μg/mL	AGS	cell viability ↓, cytotoxicity ↑, apoptosis ↑, S-phase cell cycle arrest ↓	[203]
PMMA-AA/ZnO NPs	0.0001–1 μg/mL	AGS	cell viability ↓, cytotoxicity ↑	[203]
Cur@NaCas@CaP	1–5 μg/mL	MGC-803	cell viability ↓, cytotoxicity ↑, antioxidant activity ↑	[204]
Curcumin-loaded PLGA NPs	5–40 μM	AGS	cell proliferation ↓, apoptosis ↑	[205]
Curcumin-loaded liposomes	0.78–12.5 μM	HGC-27	cytotoxicity ↑	[206]

* nd: not determined.

## Data Availability

Not applicable.

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
