# Peer review of "Curcumin-Based Nanoparticles: Advancements and Challenges in Tumor Therapy"

_pharmaceutics, 2025, doi:10.3390/pharmaceutics17010114_

Round 1
Reviewer 1 Report
Comments and Suggestions for Authors
Dear Authors,
The study entitled "Curcumin-Based Nanoparticles: Advancements and Challenges in Tumor Therapy" discusses the recent nanosized systems loaded with curcumin for the anticancer therapy. In spite of the large amount of the publications in this field, the Authors were able to demonstrate the novel review and highlight the mechanisms, as well as the results of preclinical and clinical studies.
I have several suggestions and comments, which, in my opinion, could help to improve the quality and readability of this paper.
1. P. 2, line 60, section 1 (Introduction)
I kindly recommend the Authors to add the known values of water solubility, oral bioavailability, cellular drug accumulation (if possible), and curcumin metabolic rate.
2. Section 4. Nanotechnology for curcumin delivery
Please, highlight the disadvantages or limitations associated with the curcumin-loaded nanoparticles.
3. Section 5
I kindly recommend adding the summary table with information discussed in this section, e.g. type and composition of the nanoparticles, dosage, cell lines, and the observed effects.
Author Response
Dear Authors,
The study entitled "Curcumin-Based Nanoparticles: Advancements and Challenges in Tumor Therapy" discusses the recent nanosized systems loaded with curcumin for the anticancer therapy. In spite of the large amount of the publications in this field, the Authors were able to demonstrate the novel review and highlight the mechanisms, as well as the results of preclinical and clinical studies.
I have several suggestions and comments, which, in my opinion, could help to improve the quality and readability of this paper.
We would like to sincerely thank Reviewer 1 for their thoughtful and constructive feedback. We greatly appreciate the time and effort invested in reviewing our manuscript. Below, we provide a detailed response to the points raised:
- 2, line 60, section 1 (Introduction)
I kindly recommend the Authors to add the known values of water solubility, oral bioavailability, cellular drug accumulation (if possible), and curcumin metabolic rate.
Response:
We understand the importance of providing a comprehensive overview of curcumin's properties in the introduction. To address this, we have included in the revised manuscript the requested details, specifically the known values of water solubility, and curcumins oral bioavailability, along with studies related and metabolic rate, but concerning cellular drug accumulation several parameters are taken into consideration, making it too detailed for an introduction. We would like to highlight that the manuscript already contains a dedicated section (i.e., section 3) discussing the challenges associated with curcumin.
- Section 4. Nanotechnology for curcumin delivery
Please, highlight the disadvantages or limitations associated with the curcumin-loaded nanoparticles.
Response:
We agree with the reviewer’s suggestion to highlight the disadvantages or limitations associated with curcumin-loaded nanoparticles. In response, and to ensure fluency and comprehensiveness in our review, we have included a dedicated section titled "7. Challenges in the development and clinical application of curcumin-loaded nanoparticles". This section specifically addresses the disadvantages and limitations associated with curcumin-loaded nanoparticles, as follows:
- Toxicity: Some nanoparticles may pose potential toxicity concerns, especially with certain types of materials (e.g., metal-based nanoparticles).
- Stability Issues: Curcumin-loaded nanoparticles may face challenges related to stability, including aggregation or degradation under physiological conditions, which can affect their therapeutic efficacy.
- Manufacturing and Scalability: The production of curcumin-loaded nanoparticles on a large scale for clinical use presents challenges in terms of cost, reproducibility, and quality control.
- Biodistribution and Clearance: The biodistribution of nanoparticles can vary depending on the delivery system, and clearance mechanisms may limit their prolonged therapeutic action.
These points have now been incorporated into Section 7 to give readers a balanced view of both the advantages and challenges associated with nanotechnology for curcumin delivery.
- Section 5
I kindly recommend adding the summary table with information discussed in this section, e.g. type and composition of the nanoparticles, dosage, cell lines, and the observed effects.
Response:
We appreciate the suggestion to include a summary table in this section. In response, we have added a table summarizing the key findings from the studies discussed in Section 5.
We hope that these revisions meet your expectations, and we thank you once again for your valuable feedback.

Reviewer 2 Report
Comments and Suggestions for Authors
The review provides a comprehensive overview of the recent advancement of CNPs for tumor therapy and highlights the mechanisms by which CNPs exert their anticancer. The preclinical and clinical studies are also included as well as the challenges and outlook of CNPs to clinical practice. This review is well-structured and informative, but needs to be addressed the following issues before acceptance:
1) While the challenges of curcumin clinical application are discussed, a more in-depth exploration of the limitations of current nanoparticle formulations should be included. This could involve discussing specific barriers in manufacturing, regulatory hurdles, or patient-specific factors that may influence treatment outcomes
2) Please add a summarized Table for preclinical studies related to the anticancer activity of CNPs in various types of cancers.
3) Ensure that all references are consistently formatted according to the journal's guidelines.
Author Response
The review provides a comprehensive overview of the recent advancement of CNPs for tumor therapy and highlights the mechanisms by which CNPs exert their anticancer. The preclinical and clinical studies are also included as well as the challenges and outlook of CNPs to clinical practice. This review is well-structured and informative, but needs to be addressed the following issues before acceptance:
We would like to thank the reviewer for their thoughtful and constructive feedback on our manuscript. We have carefully addressed all the points raised and made the necessary revisions. Below is a point-by-point response:
- While the challenges of curcumin clinical application are discussed, a more in-depth exploration of the limitations of current nanoparticle formulations should be included. This could involve discussing specific barriers in manufacturing, regulatory hurdles, or patient-specific factors that may influence treatment outcomes
Response:
We appreciate the reviewer’s valuable suggestion to provide a more in-depth exploration of the limitations associated with curcumin-loaded nanoparticles. To enhance the fluency and comprehensiveness of our review, we have added a dedicated section titled "7. Challenges in the Development and Clinical Application of Curcumin-Loaded Nanoparticles". This section highlights the disadvantages and limitations of these nanoparticles, offering a detailed analysis of these challenges, aligning with the reviewer's suggestion to delve deeper into the barriers to clinical application.
- Please add a summarized Table for preclinical studies related to the anticancer activity of CNPs in various types of cancers.
Response:
We have created a summarized table (Table 1) in the revised manuscript to present preclinical studies on the anticancer activity of CNPs
We hope that these revisions meet your expectations, and we thank you once again for your valuable feedback.
- Ensure that all references are consistently formatted according to the journal's guidelines.
Response:
We have thoroughly reviewed and updated all references to ensure they adhere strictly to the journal's formatting guidelines. Specifically, we have standardized citation styles, ensured correct use of italics for journal names, and verified the inclusion of DOIs where applicable.
